# Hotspots of aberrant enhancer activity punctuate the colorectal cancer epigenome

Andrea J. Cohen[1], Alina Saiakhova[1], Olivia Corradin[1,2], Jennifer M. Luppino[1], Katreya Lovrenert[1], Cynthia F. Bartels[1], James J. Morrow[1,3], Stephen C. Mack[4], Gursimran Dhillon[1], Lydia Beard[5], Lois Myeroff[5], Matthew F. Kalady[4,5,6], Joseph Willis[3,5,7], James E. Bradner[8,9], Ruth A. Keri[1,5,10], Nathan A. Berger[1,5,7], Shondra M. Pruett-Miller[11], Sanford D. Markowitz[1,5,7] & Peter C. Scacheri[1,4,5]

In addition to mutations in genes, aberrant enhancer element activity at non-coding regions of the genome is a key driver of tumorigenesis. Here, we perform epigenomic enhancer profiling of a cohort of more than forty genetically diverse human colorectal cancer (CRC) specimens. Using normal colonic crypt epithelium as a comparator, we identify enhancers with recurrently gained or lost activity across CRC specimens. Of the enhancers highly recurrently activated in CRC, most are constituents of super enhancers, are occupied by AP-1 and cohesin complex members, and originate from primed chromatin. Many activate known oncogenes, and CRC growth can be mitigated through pharmacologic inhibition or genome editing of these loci. Nearly half of all GWAS CRC risk loci co-localize to recurrently activated enhancers. These findings indicate that the CRC epigenome is defined by highly recurrent epigenetic alterations at enhancers which activate a common, aberrant transcriptional programme critical for CRC growth and survival.

[1] Department of Genetics and Genome Sciences, Case Western Reserve University School of Medicine, 10900 Euclid Ave, Cleveland, Ohio 44106, USA. [2] Whitehead Institute for Biomedical Research, 9 Cambridge Center, Cambridge, Massachusetts 02142, USA. [3] Department of Pathology, Case Western Reserve University School of Medicine, 10900 Euclid Ave, Cleveland, Ohio 44106, USA. [4] Department of Stem Cell Biology and Regenerative Medicine, Lerner Research Institute, Cleveland Clinic, 9500 Euclid Ave, Cleveland, Ohio 44195, USA. [5] Case Comprehensive Cancer Center, Case Western Reserve University, 10900 Euclid Ave, Cleveland, Ohio 44106, USA. [6] Department of Colorectal Surgery, Cleveland Clinic Foundation, 9500 Euclid Ave, Cleveland, Ohio 44195, USA. [7] Department of Medicine, University Hospitals Cleveland Medical Center, 11100 Euclid Ave, Cleveland, Ohio 44106, USA. [8] Department of Medical Oncology, Dana-Farber Cancer Institute, 450 Brookline Ave, Boston, Massachusetts 02215, USA. [9] Department of Medicine, Harvard Medical School, 25 Shattuck St, Boston, Massachusetts 02115, USA. [10] Department of Pharmacology, Case Western Reserve University School of Medicine, 10900 Euclid Ave, Cleveland, Ohio 44106, USA. [11] Genome Engineering and iPSC Center, Department of Genetics, Washington University, 4515 McKinley Building, St. Louis, Missouri 63110, USA. Correspondence and requests for materials should be addressed to P.C.S. (email: pxs183@case.edu).

The development of cancer is closely associated with the accumulation of not only oncogene and tumour suppressor mutations, but also epigenetic changes that alter chromatin structure and lead to dysregulated gene expression. In mammalian cells, active gene enhancer elements are contained within open chromatin marked with high levels of mono-methylated lysine 4 and acetylated lysine 27 on histone H3 (H3K4me1 and H3K27ac)[1,2]. We previously demonstrated that malignant transformation of colon is accompanied by widespread locus-specific gains and losses of enhancer activity, which we termed variant enhancer loci (VELs)[3]. Subsequent studies have shown that colorectal cancer (CRC) and other forms of cancer contain clusters of aberrantly active gene enhancers called super enhancers that drive dysregulated expression of oncogenes[4–6]. Additionally, both super enhancers and typical enhancers are enriched for SNPs that confer genetic predisposition to cancer[3,4,7,8]. Collectively, these studies suggest that aberrant enhancer activity is a fundamental driver of tumour formation and maintenance.

To date, a handful of different tumour types and cell lines have been molecularly profiled at the level of the enhancer epigenome. However, thorough characterizations of the enhancer epigenomes of a single type of cancer, including CRC, have been limited[9]. Additionally, because the cell type of origin for most cancers is either unknown or difficult to obtain, few studies have interrogated tumour enhancer landscapes in relation to an appropriate normal comparator. Consequently, the degree of aberrant enhancer activity in most forms of cancer remains unknown. Likewise, it is unclear whether regions of altered enhancer activity are heterogeneous across tumours of a given type or if tumours contain recurrently altered enhancers that are functionally analogous to well documented mutational hotspots[10]. The lack of a normal comparator also precludes the ability to interrogate the chromatin status of such potential hotspots before malignant transformation. Additionally, while there are strong correlations between cell type-specific enhancers and tumour risk SNPs identified through GWAS, the extent of these correlations for a given tumour type is difficult to determine without a complete reference map. It is also essential to study the epigenomes of both the normal cells and the tumour to determine the cellular context(s) in which the *cis*-regulatory SNPs likely function. Lastly, whether aberrant enhancers in cancer represent drivers of tumorigenesis or are simply bystanders accrued during malignant transformation remains to be investigated.

Here, we identified differential enhancer usage between normal colonic crypts and more than 40 CRC specimens, representing the most extensive delineation of enhancer alterations in a single type of cancer to date. We found that thousands of enhancers were activated or silenced more frequently across the CRC cell lines than expected by chance. These recurrently altered enhancers were associated with dysregulated expression of predicted target genes. Experimental manipulation of aberrantly gained enhancers, either by targeted genome editing or by pharmacologic inhibition, counteracted the associated gene overexpression, indicating a direct link between enhancer alteration and oncogenic patterns of gene misexpression. Recurrently gained enhancers frequently arise from sites of poised chromatin in normal colon tissue, overlap the majority of CRC risk loci identified by GWAS, and are commonly occupied by a set of factors including AP-1 and cohesin complexes, which may provide clues to the mechanisms underlying epigenetic dysregulation at enhancers. Approximately half of the recurrent gained VELs are present in early adenomas, suggesting that the full enhancer signature may be required for progression to full-blown CRC. We conclude that the formation of CRC is accompanied by a signature pattern of epigenetic changes at enhancer elements that contributes to tumour risk, progression, growth and survival.

## Results

**Epigenomic profiling reveals enhancer changes in CRC.** We combined 36 new and six previously published datasets[3], all uniformly processed, to obtain a set of high-resolution H3K27ac ChIP-seq profiles on seven freshly isolated purified specimens of normal colonic epithelial crypts, 31 genetically diverse CRC cell lines representing all clinical stages, and four primary colorectal tumours directly resected from patients (Supplementary Table 1). We also performed H3K4me1 ChIP-seq and DNaseI Hypersensitivity (DHS) mapping on normal colonic crypts and a subset of the CRC cell lines. We chose to use cell lines for our discovery phase analyses and tumour tissue for validation based on the knowledge that heterogeneity of tumour tissue samples and varying degrees of stromal contamination can skew genomic analyses, particularly those involving clustering, correlation and differential analyses[11]. Each sample was input matched and sequenced to a mean depth of 38.5 million uniquely aligned reads (Supplementary Data 1). A representative browser view of the H3K27ac ChIP-seq data is shown in Fig. 1a. We detected a mean of 14,406 promoters across all colon samples, defined by H3K27ac peaks (MACS $P < 1 \times 10^{-9}$, see 'Methods') located within 2 kb of transcription start sites (TSS) (Fig. 1b). The mean number of H3K27ac peaks located distal to promoters ($> 2$ kb from TSS) was 32,136. Similar results were obtained when a more conservative distance of $\leq 1$ kb from the nearest TSS was used for promoters, with only 1–5% of peaks per sample located between 1 and 2 kb of a TSS. Through overlaps of H3K27ac peaks with H3K4me1 and DHS peaks from representative samples, we determined that the majority of the distal H3K27ac peaks bear signature features of active enhancer elements. Specifically, on average 82% of H3K27ac peaks were co-enriched for H3K4me1, and 55–64% were contained within open chromatin.

To identify peaks differentially enriched for H3K27ac in each CRC sample relative to the normal colonic crypts, we used DESeq[12] and a Benjamini–Hochberg corrected $P$ value threshold of $< 0.05$ (Fig. 1c). The DESeq approach minimizes potential false positives due to discrepancies in sequence read depths. In keeping with previous terminology, we term these regions VELs. Gained VELs were defined as sites in which the H3K27ac mark was more enriched in CRC than in the normal crypts. Lost VELs were defined as sites more enriched for H3K27ac in crypts than in CRC. Exemplar loci are shown in Fig. 1d. In all cases, the percentage of gained and lost VELs within 2 kb of TSSs was far fewer than those more distal to TSSs (67–84% at distal loci, Mann–Whitney–Wilcoxon (MWW) $P < 1 \times 10^{-10}$). VELs were also enriched for distal, putative enhancer sites compared with the overall distribution of H3K27ac peaks ($\chi^2$, $P < 1 \times 10^{-30}$), suggesting distal regulatory elements may be particularly prone to aberrant changes in cancer. Gained VELs showed concordant increases in H3K4me1 levels and were often located in open chromatin sensitive to DNaseI digestion (Fig. 1d, Supplementary Fig. 1A,B). Lost VELs showed concordant decreases in H3K4me1 and were located in closed chromatin (Fig. 1d, Supplementary Fig. 1A,C). We assigned VELs to their putative target genes using PreSTIGE, an experimentally validated computational method that predicts enhancer–gene interactions based on integrated enhancer profiles and gene expression data[13–15]. Relative to normal colon, genes associated with gained VELs showed elevated expression in CRC, while lost VEL genes were broadly repressed, and the magnitude of the change in expression positively correlated with the number of VELs per gene (Fig. 1e).

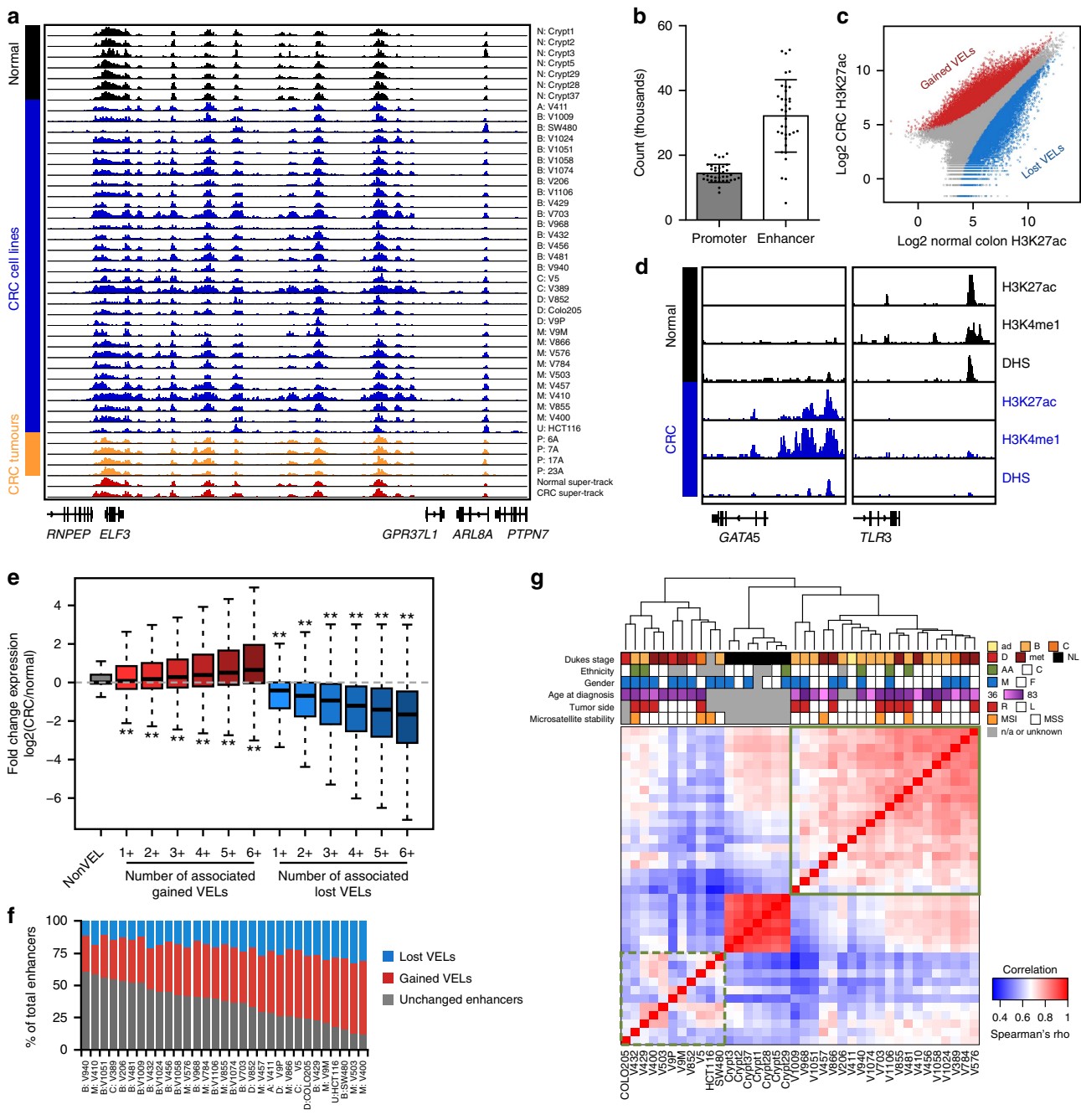

**Figure 1 | Epigenetic alterations at enhancers distinguish CRC from normal colon.** (**a**) Normalized H3K27ac ChIP-seq tracks of all CRC cell line, colon crypt, and tumour samples at a representative locus (hg19, chr1:201,960,000–202,129,000). Super-tracks (red) correspond to median binned signal of all normal crypts or all CRC cell lines. Each track is labelled with the sample type and name. (**b**) Average number of H3K27ac ChIP-seq peaks that overlap promoters ($\leq$2 kb from TSS; grey) and putative enhancers ($>$2 kb from TSS; white). Dots represent counts in each of the 31 CRC cell lines and four normal colon samples, bars represent mean $\pm$ s.d. (**c**) Relative enrichment of H3K27ac ChIP-seq signal in a representative CRC cell line (V410) compared with normal colon crypts. Peaks differentially enriched between normal and CRC (Benjamini–Hochberg corrected $P<0.05$) are shown in red and blue. (**d**) H3K27ac and H3K4me1 ChIP-seq and DHS tracks at exemplar gained (left) and lost (right) VELs in a representative CRC cell line (V410; blue) compared with a normal colon sample (black). (**e**) Fold change of expression of all genes associated with gained (red) and lost (blue) VELs, and genes not associated with VELs (nonVEL; grey). **MWW $P<1\times10^{-99}$ versus non-VEL genes. Box boundaries represent 1st and 3rd quartiles, middle line represents median, and whiskers extend to the nearer of the data extremes or 1.5 times the IQR. (**f**) Percentage of lost (blue), gained (red), and unchanged (grey) enhancers detected in each CRC cell line. (**g**) Unsupervised clustering of all pairwise correlations of enhancer RPKMs for CRC cell lines and normal colon samples. Colour bars (middle) indicate clinical and molecular characteristics of each sample. Green boxes indicate samples that are more (solid line) and less (dashed line) similar to one another and to crypts. A,ad, adenoma; B–D, Dukes stage B–D; M,met, metastasis; N,NL, normal; U, unknown; P, primary tumour; AA, African American; C, Caucasian; M, male; F, female; R, right; L, left; MSI, microsatellite instable; MSS, microsatellite stable.

We determined that our cohort of 31 CRC samples captured 89% of all possible lost VELs and 80% of all possible gained VELs in CRC, indicating that our VEL detection approached 'saturation' (Supplementary Fig. 1D). We predicted that each additional cell line sequenced after 31 samples would add <1% to the total number of unique VELs identified. The percentage of VELs in each CRC cell line relative to all enhancers (gained, lost, and unchanged) varied widely among CRC samples, from 39 to 88% (Fig. 1f). Unsupervised clustering showed that the enhancer epigenomes of the normal crypts were highly distinct from those of CRC. CRC cell lines formed two clusters. Members of one cluster were considerably more closely correlated to one another (median Spearman's correlation $\rho$ 0.73) and to the normal crypts (median $\rho$ 0.65) than members of the other cluster (median $\rho$ 0.61 and 0.54, respectively) (Fig. 1g, compare green boxes). The more correlated, 'crypt-like' cluster was more enriched for early stage CRCs than the less crypt-like cluster (Z test for two-proportions $P < 0.05$). The less crypt-like cluster was also male biased (Z test $P < 0.05$). The clusters failed to correlate with microsatellite instability status, tumour side, patient ethnicity, or age at diagnosis. We conclude that colon cancer progression is accompanied by extensive remodelling of the enhancer epigenome, with early stage tumours generally preserving a considerable portion of the normal enhancer epigenome and appearing more 'crypt-like' than late stage tumours that have undergone a more dramatic shift.

**Highly recurrent VELs dysregulate key cancer genes.** Upon visual inspection of ChIP-seq profiles, we noticed 'hotspots', that is, genomic intervals harbouring VELs shared by the vast majority or all of the CRC samples, suggesting they were positively selected during malignant transformation of the colon. An example is shown in Fig. 2a, where multiple gained VELs at the FOXQ1 locus are evident in nearly all CRC samples. To systematically assess VEL recurrence, we used permutation analyses to identify VELs common among a greater proportion of CRC samples than expected by random chance at various stringent false discovery rates (Fig. 2b). Enhancers gained in 10 or more CRC lines (G10+) or lost in 14 or more CRC lines (L14+) were significantly recurrent (permutation $P < 0.001$, FDR < 0.05; Supplementary Data 2 and 3). We detected 75 gained VELs and 67 lost VELs common to all 31 CRC cell lines (FDR < 0.0001). More than 90% of the most highly recurrent VELs (present in 30 or all 31 of the CRC lines) validated in primary tumours and are therefore unlikely to be cell culture artifacts (Fig. 2c). Expression of genes associated with recurrent gained VELs was elevated across primary CRCs relative to normal, while lost VEL gene expression was repressed (Fig. 2d, MWW $P < 1 \times 10^{-43}$). Moreover, dysregulated genes associated with recurrent VELs were more likely to validate as dysregulated in patient tumours than misexpressed genes not associated with VELs ($\chi^2$, $P < 1 \times 10^{-10}$). Additionally, genes associated with the most recurrent VELs were more dysregulated than genes associated with less recurrent VELs (MWW $P < 1 \times 10^{-31}$; Supplementary Fig. 2A). To directly characterize the regulatory effects of VELs, we performed CRISPR/Cas9-mediated disruption of three recurrently gained enhancers predicted to upregulate PHLDA1 in HCT116 cells (Fig. 2e). Compared with the unedited parental cell line, PHLDA1 levels were reduced by more than 60% in each of the three the cell lines containing the edited enhancers (Fig. 2f). As a negative control, we also used CRISPR/Cas9 to disrupt four sites at the MYC locus that had relatively weak signal in HCT116 cells, but were identified as robustly gained VELs in other CRC cell lines (Supplementary Fig. 2B). The disruption of these four sites did not significantly impact MYC levels (Supplementary

Fig. 2C). Together these results suggest that VELs have a significant regulatory effect on the predicted target genes.

Network analysis of gene ontology terms enriched among recurrent gained VEL genes revealed commonalities including embryogenesis, angiogenesis, hormone secretion, DNA replication, small molecule transport and drug metabolism (Fig. 2g). Gene ontology terms enriched among lost VEL genes included small RNA regulation, ion homoeostasis, ion transport, drug metabolism and cell transporter activity. Genes associated with the most common VELs included novel genes and several known oncogenes and tumour suppressors implicated in CRC as well as other forms of cancer, including MYC[10,16], BMP4 (refs 17,18), PHLDA1 (refs 19–22), SOX9 (refs 23–25) and TRIB3 (refs 26,27). Several of the recurrent VEL genes have been shown through functional studies to enhance tumorgenicity in CRC. For example, overexpression of FOXQ1 has been shown to increase CRC tumour growth in mice[28]. Reduced expression of recurrent lost VEL genes, E2F2 and SIRT6, is associated with increased tumour growth in CRC[28–31]. Overall, these findings are consistent with and extend our previous studies[3] indicating that the CRC epigenome is defined by a signature set of highly recurrent epigenetic alterations at enhancer elements, or 'hotspots'. The highly significant recurrence of these enhancer aberrations suggest that these loci may be under positive selection for acquisition of enhancer function. This is further evidenced by the association of these hotspots with a common aberrant transcriptional programme defined by a broad range of cellular processes that are frequent hallmarks of colorectal and other types of cancer.

**AP-1 and cohesin factors enriched at recurrent gained VELs.** Recurrent gained VELs often contained exceptionally high levels of H3K27ac and formed clusters of individual enhancers, reminiscent of super-enhancers. These observations prompted us to systematically investigate the relationship between VELs and super-enhancers. To define super-enhancers in each sample, we implemented the ROSE script on H3K27ac ChIP-seq signals (Supplementary Fig. 3A), and then determined the fraction of gained and lost VELs that were constituents of super-enhancers as a function of VEL recurrence rate. Twenty-four to 44% of lost VELs were constituents of super-enhancers in normal colon crypt, and this percentage did not scale with the lost VEL recurrence rate (Supplementary Fig. 3B). In marked contrast, the proportion of gained VELs that were super-enhancer constituents progressively increased with VEL recurrence. Of the most highly recurrent gained VELs, 96% were constituents of super-enhancers, compared with <20% of gained VELs unique to a single CRC line.

We next set out to identify transcription factors and other proteins that occupy the recurrent gained VELs and potentially activate expression of the associated genes. We began by identifying transcription factor motifs enriched in recurrent gained VELs compared with unique gained VELs. Remarkably, five of the top six most enriched factors were members of the AP-1 complex (HOMER $P < 1 \times 10^{-300}$; Supplementary Data 4). AP-1 factors mediate cell state transitions during development and in response to myriad environmental cues[32] that are often exploited by cancer cells for growth and survival. We next integrated the CRC VEL profiles with publicly available ChIP-seq data of more than 400 proteins in CRC cells[33], looking for factors that showed the greatest increase of enrichment with VEL recurrence (Supplementary Fig. 4). Consistent with the motif search, AP-1 factors ranked among the top, including JUND and JUN in the top 2%, and ATF2 in the top 6%. Three core components of the cohesin complex, RAD21, SMC1A and SMC3,

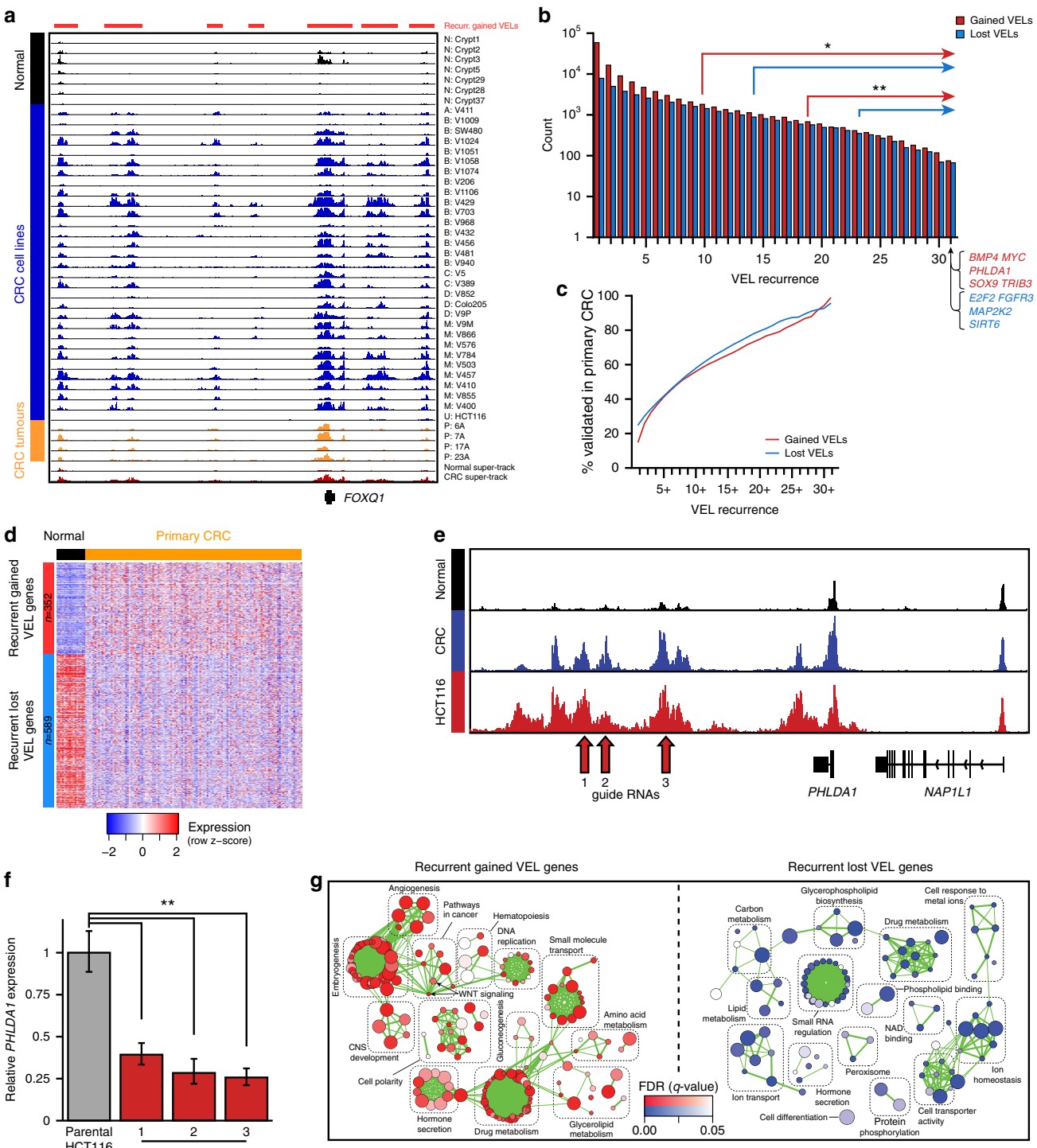

**Figure 2 | VELs are recurrent and associated with altered gene expression.** (**a**) Normalized H3K27ac ChIP-seq tracks at the *FOXQ1* locus. Red bars (top) denote gained VELs present in 10 or more CRC cell lines (G10 +). Colours and labels as in Fig. 1a. Track heights are all equal. (**b**) Quantification of gained (red) and lost (blue) VEL recurrence. Arrows correspond to two different thresholds of recurrence: Permutation *$P < 0.001$, FDR $< 0.05$; **$P < 0.001$, FDR $< 1 \times 10^{-5}$. Examples of genes associated with VELs present in all 31 CRC lines are shown at lower right. (**c**) Per cent of VELs validated in primary tumours at each recurrence rate. (**d**) Expression levels of recurrent gained and lost VEL genes in normal colon tissue and CRC patient tumours. (**e**) Normalized H3K27ac ChIP-seq track for HCT116 compared with CRC and normal colon H3K27ac super-tracks (median binned signal of all normal crypts or all CRC cell lines) at the *PHLDA1* locus. Guide RNA target sites for the three CRISPR–Cas9 edits performed are indicated by arrows. (**f**) Barplots indicate *PHLDA1* RNA levels in the parental HCT116 cell line and cells disrupted by CRISPR/Cas9 at each of the sites indicated in Fig. 2e (relative to *GAPDH*, mean ± 95% confidence interval of quadruplicates). **$T$-test $P < 0.0001$. (**g**) Gene ontology networks for genes associated with recurrent gained (G10 +, left) and lost (L14 +, right) VELs. Circles represent gene ontology categories with size proportional to number of genes and coloured based on enrichment FDR. Green lines connect categories with overlapping gene sets with the line weight proportional to the degree of overlap.

as well as the cohesin loading factor, NIPBL, were also among the top ranking factors (rank > 96th percentile). The cohesin complex facilitates physical interactions among individual constituents of super-enhancers and their target genes[34,35]. Factors ranking highly in this analysis also included several that were themselves targets of highly recurrent VELs, such as *MYC*, *ETS2* and *CDX2* (rank ~99th, 97th and 94th percentile, respectively), indicative of autoregulatory loops driving expression of these factors, as has been previously demonstrated for super enhancer-associated genes[9,36].

**Recurrent gained VELs are hotspots for CRC GWAS risk loci**. We next sought genetic evidence that recurrent VELs are relevant to CRC by intersecting various categories of VELs and enhancers with 75 loci associated with genetic predisposition to CRC through GWAS (Supplementary Data 5)[37]. Consistent with previous results, the majority of risk loci (76.0%; risk SNP and SNPs in tight linkage disequilibrium, see 'Methods') reside in a putative H3K27ac enhancer in crypts or CRC[3,7]. Surprisingly, two-thirds of these (48% of all CRC risk loci) mapped to recurrent gained VELs, most of which were highly recurrent, that is, common to 19 or more CRC samples (Fig. 3a). Through variant set enrichment (VSE) analyses[3,38] we confirmed that CRC risk locus variants were most enriched in highly recurrent gained VELs, and that this enrichment was specific for recurrent gained VELs and not recurrent lost VELs or other genomic features such as 5′UTRs, exons or introns (Fig. 3b). Promoter enrichment was borderline significant, but not to the degree of recurrent or highly recurrent gained VELs. We highlight an exemplar locus containing two independent disease association signals that lie in recurrent gained VELs downstream of *DUSP10* (Fig. 3c). The full list of risk loci that overlapped recurrent gained VELs and their putative target genes is shown in Fig. 3d. We also determined that the proportion of VELs containing CRC risk SNPs scaled with the recurrence of the VELs, such that highly recurrent VELs common to 19 or more CRC lines were 7.8-fold more likely to overlap a risk locus than VELs unique to a single CRC line (Fig. 3e, Supplementary Fig. 5). The genomic convergence between recurrent VELs and CRC risk loci provides genetic evidence that the recurrent gained VELs are relevant in colorectal tumorigenesis. Furthermore, the results imply that a considerable fraction of the CRC GWAS signals are *cis*-regulatory variants active only in the tumour, as has been previously proposed[7,39]. This is consistent with either of two possibilities: that the CRC risk SNPs promote aberrant gain of enhancer function, or that the SNPs alter the activity of the enhancer following its acquisition in cancer.

**A subset of recurrent gained VELs present in colon adenomas**. We next set out to determine if the recurrent gained VELs arise before or after malignant transformation. We isolated two early adenomas from a patient clinically diagnosed with familial adenomatous polyposis and performed H3K27ac ChIP-seq studies. We then identified VELs in the adenomas and compared them to the set of recurrent gained VELs. Of the gained VELs recurrent in 19 or more CRC cell lines (G19+), 38% were also identified as VELs in at least one adenoma (Fig. 4a). Of VELs identified in 30 or more CRC cell lines, 70% were detected in the adenoma samples. We next compared the H3K27ac ChIP-seq signal strength of the most highly recurrent gained VELs (G30+) in the CRC cell lines versus the adenoma samples. This analysis revealed a population of VELs with relatively equal signal strength in both the adenomas and the CRC samples (Fig. 4b), and a population where the relative VEL signal was at least two-fold higher in the CRC samples versus the adenomas (Fig. 4b,c,

left). Genes associated with the more CRC-specific VELs included several well known oncogenes in CRC, including MYC and TCF7L2 (Supplementary Data 6). Genes associated with the recurrent gained VELs shared between CRC and the adenomas included FOXQ1 and RASGRF1 (Fig. 4c, right). The results suggest that during the stepwise progression of CRC, the crypt to early adenoma transition is accompanied by acquisition of a subset of the recurrent gained VEL signature, and that the remaining signature VELs likely arise later, during the adenoma to carcinoma transition.

**Recurrent VELs originate in primed and poised chromatin**. Cell state transitions during embryonic development are mediated by dynamic and coordinated changes in enhancer activity that ensure proper spatiotemporal gene expression[40,41]. These changes involve commissioning and decommissioning of enhancers, as well as dynamic switching of primed (H3K4me1) and poised (H3K4me1 and H3K27me3) chromatin to the active state (H3K4me1 and H3K27ac). Dynamic switching between primed and active chromatin states also occurs in terminally differentiated cells in response to both intrinsic and extrinsic stimuli[42,43]. Based on the notion that malignant transformation fundamentally represents a major transition in cell state, and that tumour expression programs are often responsive to microenvironmental cues, we investigated the chromatin status of gained VELs before malignant transformation, through analysis of H3K4me1 and H3K27me3 ChIP-seq data from the normal colon crypts. Of non-recurrent gained VELs (VELs unique to one CRC line), 20% contained significant levels of H3K4me1 in the normal crypts and were considered primed (Fig. 5a). Less than 2% contained both H3K4me1 and H3K27me3 and were considered poised. The majority gained both H3K4me1 and H3K27ac marks and appear to be newly commissioned in CRC, lacking both H3K4me1 and H3K27ac in the normal and gaining both of these marks in CRC. Strikingly, far fewer of the recurrent VELs were determined to be newly commissioned enhancers in CRC, with 64–67% defined as primed in normal and 13–24% as poised in normal ($\chi^2$, $P < 0.0005$) (Fig. 5a). Exemplar VELs that switched to active in CRC from the poised state in normal colon are shown (Fig. 5b). The findings indicate that most of the recurrent VELs were not newly commissioned, but rather that they existed within the normal crypts. This suggests a reawakening of developmental or environmentally responsive enhancers is one mechanism for recurrent gain of enhancer activity in CRC.

**Recurrent gained VEL genes may represent CRC dependencies**. While some of these VELs or their associated genes, could be 'markers' of CRC and not themselves drivers of tumorigenesis, based on the preceding results we hypothesized that a subset have a direct role in establishing or maintaining the CRC phenotype. Many of the recurrent gained VELs were super-enhancers acquired in CRC. A handful of super enhancer-associated genes, including *MYC* and *OCA-B*, have been shown to be 'dependency' genes in multiple myeloma, and these genes are often selectively downregulated in response to BET inhibition[5,6,44,45]. This led us to hypothesize that genes associated with highly recurrent gained VELs may indeed be CRC 'dependency' genes that are similarly amenable to downregulation in response to pharmacologic BET inhibition. We therefore used the BRD4 inhibitor, JQ1 (refs 44,46), as a tool to assess the functional role of the gene set demarcated by recurrent gained VELs. We tested 20 CRC cell lines in culture, which were all relatively sensitive to JQ1, showing a broad range of IC50s ranging from 100 to 3,800 nM (Fig. 6a,b). These IC50s are below the range reported for JQ1-resistant

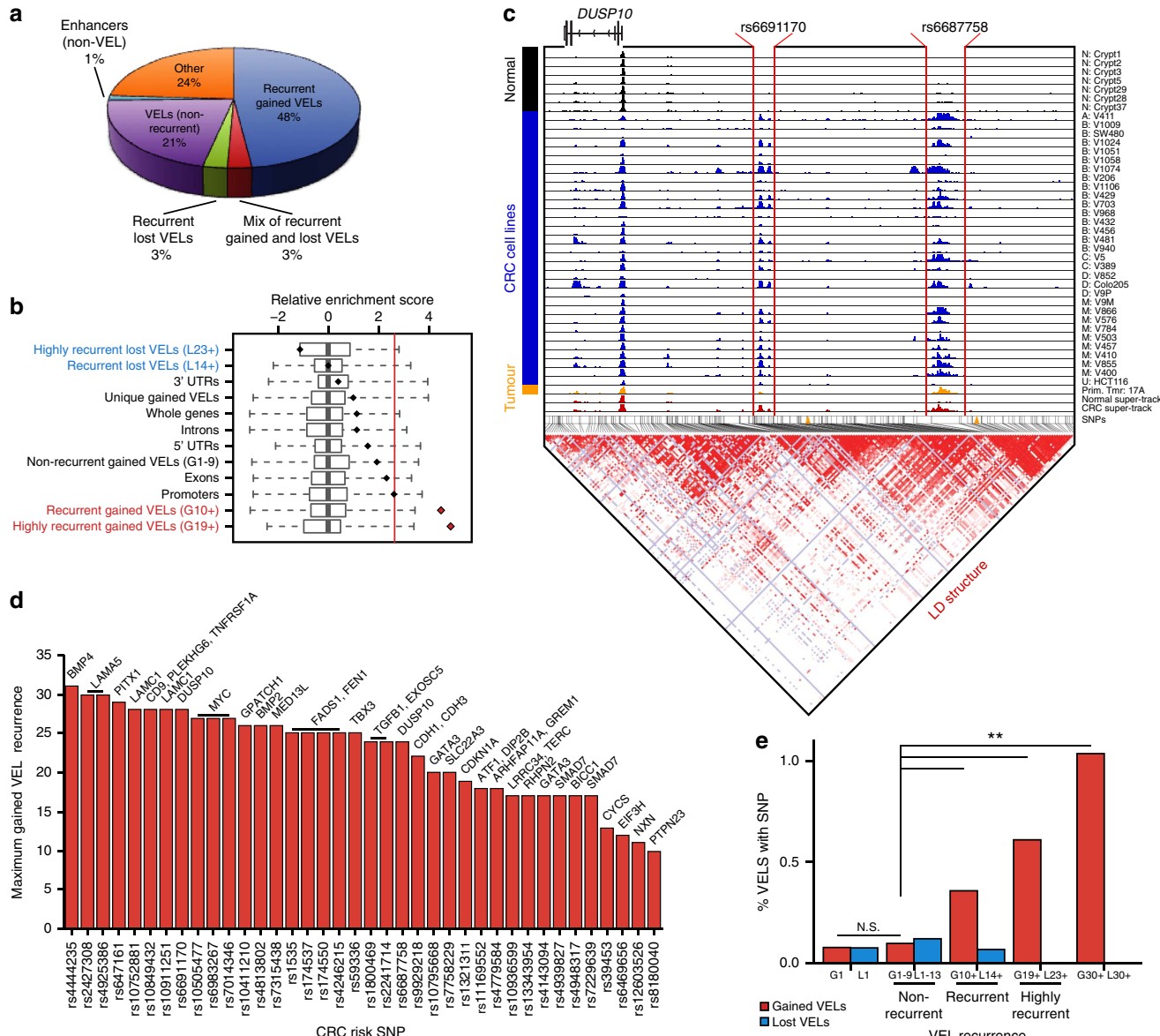

**Figure 3 | CRC risk SNPs lie in highly recurrent VELs.** (**a**) Co-localization of CRC risk SNPs and recurrent gained and lost VELs, non-recurrent VELs, and non-VEL enhancers. (**b**) Variant set enrichment (VSE) analysis showing the degree of enrichment of CRC risk SNPs (diamonds) in various genomic features relative to random SNP sets (boxplots). Red line indicates significance threshold (Bonferroni corrected $P < 0.05$). (**c**) Normalized H3K27ac ChIP-seq tracks in normal crypt and CRC at the DUSP10 locus. SNPs are represented by vertical black lines, at middle, and linked to haplotype block structure, shown below. Red lines denote the genomic locations of recurrent gained VELs which colocalize with two distinct CRC risk loci containing the lead GWAS risk SNPs rs6691170 and rs6687758 (orange arrowheads). Each track is labelled with the sample type and name of CRC cell line, crypt, or primary tumor sample. N, normal; A, adenoma; B–D, Duke's stage B–D; M, metastasis; U, unknown. Super-tracks in red correspond to median binned signal of all normal crypts or all CRC cell lines. (**d**) Maximum gained VEL recurrence at each recurrent gained VEL CRC risk locus. The lead SNP at each risk locus is shown on the x axis. Putative target genes are shown at the top of each bar. (**e**) Percentage of gained (red) and lost (blue) VELs that contain a risk SNP (lead or LD). G10+, G19+ and G30+ correspond to gained VELs recurrent in at least 10, 19 and 30 lines, respectively. L14+, L23+, L30+ correspond to lost VELs recurrent in at least 14, 23 or 30 lines, respectively. **$\chi^2$ $P < 1 \times 10^{-5}$.

cancers of various origins[47–49]. Dose–response curves for one of the most and least responsive CRC cell lines are shown in Fig. 6a. Flow cytometry analyses indicated that CRC cells treated with JQ1 underwent both cell cycle arrest and apoptosis (Supplementary Fig. 6), consistent with responses observed in other cancers[45,46]. We next tested JQ1 efficacy in mouse xenograft models of three CRC cell lines that showed variable responses *in vitro*. JQ1 had similar effects in all three xenograft models, significantly slowing tumour growth (Fig. 6c, Supplementary Fig. 7A). It is unsurprising that the xenografts

responded similarly to JQ1, given that the previously reported serum concentrations for the dosing regimen used far exceed the IC50s calculated *in vitro*[44]. To evaluate whether the growth-inhibiting effects of JQ1 were associated with a specific transcriptional response, we performed transcriptomic analysis before and after treatment with JQ1. We tested two highly sensitive, two moderately sensitive, and two less sensitive CRC cell lines at four time points after JQ1 treatment (0.5, 1, 6, 24 h) as well as untreated controls. We then analysed the transcriptional response of genes associated with gained VELs and recurrent

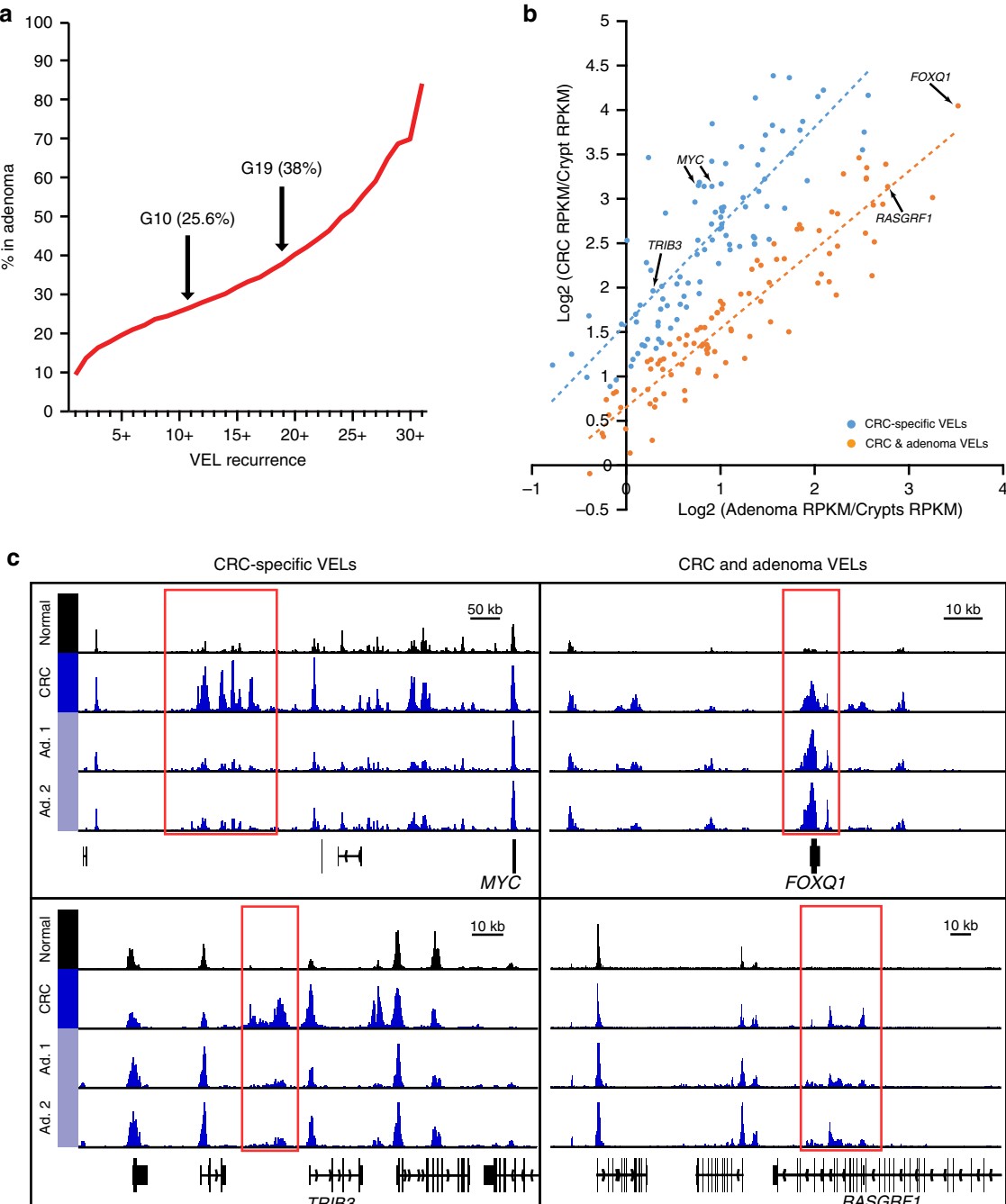

**Figure 4 | Recurrent gained VELs in adenomas.** (**a**) Per cent of gained VELs identified in primary adenoma samples at each recurrence. (**b**) Scatter plot comparing H3K27ac ChIP-seq signals at G30+ VELs in adenomas (*x* axis) versus CRC cell lines (*y* axis) relative to normal colon crypt. The linear regression line for each set of enhancers is shown (dotted lines). (**c**) Normalized H3K27ac ChIP-seq tracks for two adenomas compared with CRC and normal colon H3K27ac super-tracks (median binned signal of all normal crypts or all CRC cell lines) at four loci representative of recurrent gained VELs in the CRC cell lines that are absent (left) and present (right) in adenomas.

gained VELs relative to all expressed genes as controls. Given that genes associated with multiple gained VELs were particularly upregulated in CRC relative to normal (Fig. 1e), we also quantified the expression of these 'clustered VEL' genes following JQ1 treatment. At the six-hour time point, gained VEL genes were significantly reduced in expression in all but the two least sensitive CRC cell lines (Fig. 6d, Supplementary Fig. 7B). Recurrent gained VELs showed an even greater degree of downregulation, with genes associated with multiple recurrent gained VELs showing the most dramatic response.

Thus, recurrent gained VELs mark a set of genes that are specifically downregulated following JQ1 treatment. The response of the recurrent VEL genes was markedly attenuated in two less sensitive CRC lines compared with the two most sensitive CRC lines (Fig. 6d, Supplementary Fig. 7B). We next analysed whether the difference in the response was related to the specific genes targeted by recurrent VELs in the most and least sensitive lines, or the degree of responsiveness of the same gene targets. We found that 89–94% of the highly recurrent gained VEL genes were shared between the two most and two least sensitive CRC lines,

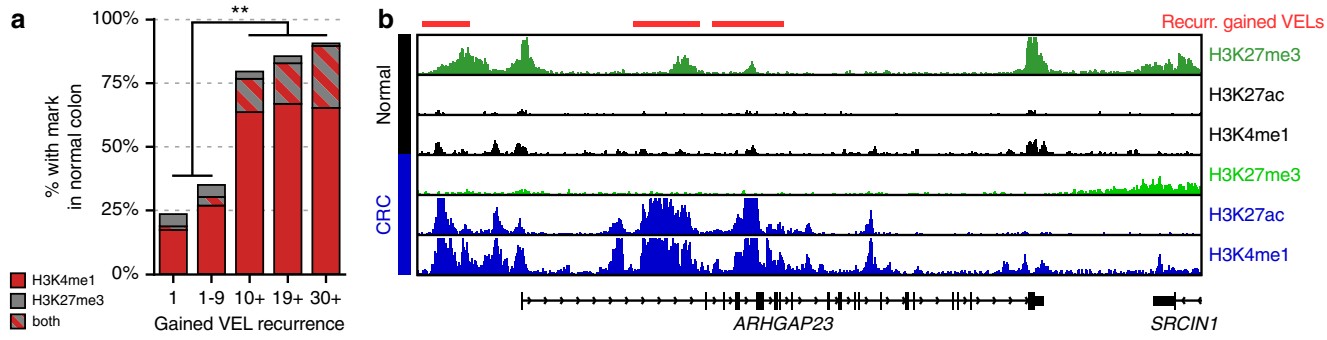

**Figure 5 | Gained enhancer loci show chromatin marks of poised and primed enhancers in normal colon.** (**a**) Bar plot of proportions of gained VELs that in normal crypt were enriched for H3K4me1, H3K27me3, or both marks. **$\chi^2$ $P < 0.0005$. (**b**) Normalized H3K27me3, H3K27ac and H3K4me1 ChIP-seq profiles at the ARHGAP23 locus. Red bars (top) denote recurrent gained VELs present in 10 or more CRC cell lines (G10 + ).

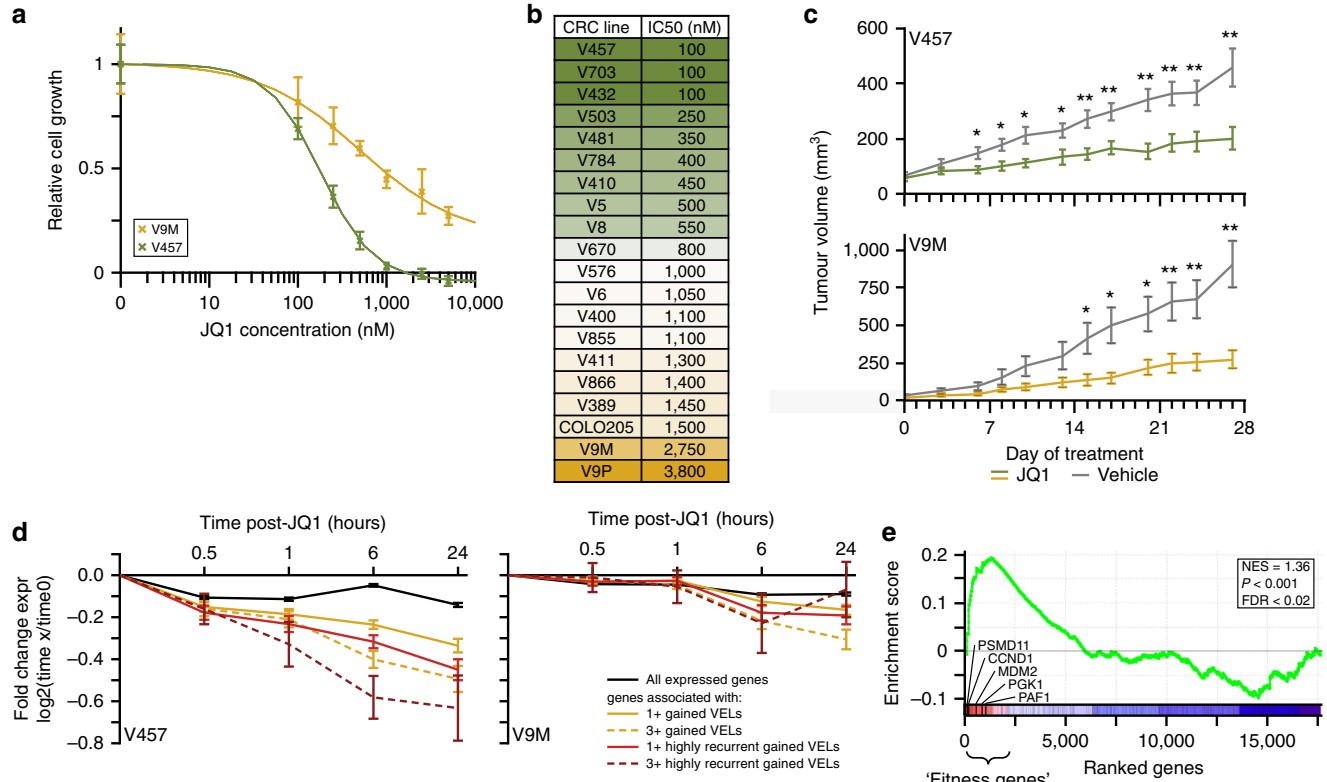

**Figure 6 | Recurrent VEL genes maintain CRC phenotype.** (**a**) Growth-inhibition curves for a relatively sensitive (green) and a less sensitive (yellow) CRC cell line treated with JQ1 for three days (relative to day 0; mean ± s.d.). (**b**) JQ1 doses required for 50% inhibition ($IC_{50}$) for all 20 CRC lines assayed. (**c**) Tumor volume in mouse xenograft models treated with 25 mg kg$^{-1}$ BID JQ1 (green/yellow) or DMSO control (grey) in a relatively sensitive (V457) and less sensitive (V9M) CRC line (mean ± s.d.). *T*-test *$P < 0.05$, **$P < 0.01$; $n = 10$ per group. (**d**) Line plots depicting fold change of expression (JQ1 treated/untreated) of various gene sets at indicated time points of JQ1 treatment (mean ± s.e.m.). (**e**) Gene set enrichment analysis (GSEA) displaying enrichment of recurrent gained VEL genes among high-confidence fitness genes from the Toronto KnockOut library in HCT116 CRC cells (NES, normalized enrichment score).

however the response of these shared genes was far greater in the two sensitive lines ($P < 0.05$). We conclude that the phenotype of growth inhibition by JQ1 is associated with recurrent VEL gene response, and not necessarily differences in the targeted gene sets between sensitivity groups.

Lastly, we tested whether individual genes associated with recurrent gained VELs might represent cancer dependencies. We took advantage of publicly available data from a recent study identifying 'fitness genes' in the CRC cell line HCT116 through CRISPR–Cas mediated knockout of 17,661 protein-coding genes[50]. Using gene set enrichment analysis (GSEA), we found a robust correlation between highly ranked fitness genes and

genes associated with the recurrent gained VELs (Fig. 6e), indicating that HCT116 cells rely on sustained expression of several of the recurrent gained VEL genes. Together with the BET inhibitor studies, these results indicate that CRC tumours are dependent on recurrent VEL genes both globally and on an individual gene basis.

## Discussion

The classic genetic model of CRC tumorigenesis, or the 'Vogelgram', states that mutation of APC initiates the conversion of normal colon epithelium to the early adenoma stage[51].

The progression from adenoma to frank carcinoma is accompanied by additional mutations in other genes including *KRAS*, *SMAD4*, *PIK3CA* and *TP53* (ref. 52). This model is well supported by exome sequencing studies showing that these genes lie in mutational hotspots and myriad functional studies support their role in malignancy. The results presented here indicate that recurrent oncogenic events are not limited to the CRC genome, but that there are also hotspots of frequent epigenetic dysregulation at enhancer elements. We found thousands of VELs that were more recurrent than expected for passenger-like events, suggesting they were under strong positive selection during the process of tumorigenesis. Additionally, similar to recurrent DNA mutations, the recurrently altered enhancer elements deregulate potential tumour suppressors and oncogenes. Interestingly, the number of recurrent enhancer alterations dwarfs that of recurrent mutational events by as much as 10-fold. Collectively, these findings indicate that epigenomic enhancer dysregulation occurs in parallel to the well described DNA mutations that occur at canonical proto-oncogenes and tumour suppressors during malignant transformation of colon crypt cells into CRC.

We provide several lines of evidence that many of the recurrently acquired enhancer changes in CRC are functional and relevant to the CRC phenotype. First, through integration with transcriptome data from both CRC cell lines and primary tumours, we find that the recurrently acquired enhancers activate genes that are commonly elevated in CRC, suggesting VEL acquisition is a major mechanism by which tumour cells alter expression to gain a growth or survival advantage. Several of these genes are known oncogenes previously implicated in CRC by TCGA and others. This includes MYC, the primary effector of Wnt-β catenin pathway activity in CRC[16] and a central 'node' connecting dysregulation of Wnt-β catenin and other signalling pathways to downstream patterns of misexpression[10]. Others genes associated with recurrent VELs are novel, and by association may also be important in CRC pathogenesis. Second, we demonstrate a strong convergence between the recurrently acquired enhancers and genetic loci associated with risk to CRC through GWAS, with recurrent gained VELs far more likely to harbour a GWAS risk SNP than non-recurrent VELs. Third, we demonstrate that the growth of CRC cells can be mitigated by targeted knockout of individual genes activated by the recurrent VELs, or with a BET inhibitor that selectively and potently suppresses genes associated with the recurrently acquired VELs. Collectively, these findings indicate that the recurrently acquired enhancers drive a specific transcriptional programme that both specifies and maintains the CRC phenotype.

What drives formation of the recurrent VELs and why these specific enhancers are recurrently activated remain outstanding questions. One possibility is that VELs are formed through somatic mutations that introduce transcription factor binding sites, similar to somatic mutations that result in recruitment of MYB to the *TAL1* enhancer previously described in T-cell acute lymphoblastic leukaemia[53]. Copy number alterations are another possibility[54], although it is worth noting that ChIP-seq peaks in our study were input-normalized, and therefore enhancer signals derived from copy number alterations are most likely excluded from our analyses. Moreover, given that the frequency of recurrent VELs exceeds that of all but the most common mutations in CRC, DNA variation alone cannot fully account for all recurrent VELs. Integration of somatic mutations derived from whole genome sequencing of matched tumour/normal pairs with the enhancer landscapes presented here could help reveal any potential interplay between the CRC genome and enhancer epigenome. Indeed, functional data annotating the non-coding

genome, such as that presented in this study, will be critical for the interpretation of WGS data in CRC and other types of cancer. An alternative hypothesis is that recurrent VELs form downstream of primary mutational events which deregulate a common signalling mechanism, such as Wnt/β-catenin. However, we found that only a subset of the recurrent VELs are present at the adenoma stage, suggesting that additional events beyond deregulated Wnt/β-catenin signalling are required for establishment of the full VEL signature and transition to the carcinoma stage. Future studies designed to profile the enhancer epigenome in colon cells immediately following complete loss of the APC gene and after each of the subsequent mutational events that accompany the stepwise formation of CRC could help shed light on these possible mechanisms.

Lastly, while there is a growing body of evidence implicating super-enhancer function in malignancy, the concept that super enhancers represent a distinct class of enhancers which constitute a new regulatory paradigm distinct from typical enhancers remains controversial[55]. The gained and lost VELs presented here were identified by an unbiased survey of the CRC enhancer epigenome and evaluated using metrics that were, by design, analogous to accepted methods for identifying likely driver DNA mutations, such as recurrence, association with known cancer genes and oncogenic pathways, and impact on gene expression. Regions containing clusters of enhancers repeatedly emerged, throughout our analyses. Recurrent gained VELs often occurred in clusters and in fact, the most recurrent gained VELs were almost universally constituents of super-enhancers. The enhancer landscape around known oncogenes was often densely populated with gained VELs, such as shown at the FOXQ1 locus (Fig. 2a). Additionally, genes associated with multiple gained VELs were not only more upregulated in CRC than genes associated with a single VEL, they were also more responsive to inhibition by JQ1. Thus, even though our method of identifying VELs made no prior assumptions about genomic distribution, our findings converge with several others[4,54,56–58] to suggest that clusters of recurrent enhancer activation have disproportionately strong effects on oncogenesis.

## Methods

**Cell culture and tissue samples.** VACO cell lines were previously derived from colorectal tumour specimens and cultured as described[59]. Briefly, VACO cell line V9M was grown in MEM media (Gibco, cat. 10370-021) supplemented with 8% fetal bovine serum (Hyclone, cat. SH30071.03), 2 mm L-glutamine (Gibco, cat. 25030-081), and 50 μg ml$^{-1}$ gentamycin (Gibco, cat. 15750-060). HCT-116 and VACO cell lines V5, V8 and V9P were grown in MEM media supplemented with 8% heat-inactivated Cosmic calf serum (Hyclone, cat. SH30087.03), 2 mm L-glutamine, and 50 μg ml$^{-1}$ gentamycin. All other VACO cell lines were grown in MEM media supplemented with 2% fetal bovine serum, 2 mM L-glutamine, 1 μg ml$^{-1}$ bovine insulin, 0.55 μg ml$^{-1}$ human transferrin, 0.05 nM sodium selenite (ITS; Fisher, cat. MT-25-800-CR), 50 μg ml$^{-1}$ gentamycin, and 1 μg ml$^{-1}$ hydrocortisone (MEM-2+). SW480 was also grown in MEM-2+ media. COLO205 was cultured in RPMI-1640 (Gibco, cat. 11875-093) supplemented with 10% fetal bovine serum, 2 mM L-glutamine and 1% penicillin/streptomycin (Gibco, cat. 15140-122). Crypt epithelial cells were isolated from normal colon surgical specimens by EDTA and mechanical dissociation, as previously described[3]. Primary CRC tumours (n = 4) were grossly dissected from colon freshly resected from four different individuals. Two adenomas, each 2–4 mm diameter, were grossly dissected from colon freshly resected from a patient with familial adenomatous polyposis. All human tissue samples were obtained in accordance with protocols approved by the Institutional Review Boards of the Case Western Reserve University Human Research Protection Program. Patient informed consent was provided for use of human tissue sample in research.

**ChIP-seq and DNase-seq.** DNase-seq was performed as previously described[60]. ChIPs were performed from 5 to 10 million cross-linked cells and sequencing libraries were prepared as previously described[61]. ChIP from frozen tumours and adenomas was prepared as previously described[62]. ChIPs were performed using 8 μg of the following antibodies: rabbit anti-H3K4me1 (Abcam #8895), rabbit anti-H3K27ac (Abcam #4729) and H3K27me3 (Abcam #6002). ChIP-seq libraries were sequenced on an Illumina HiSeq 2000 or 2500 platform at the Case Western

Reserve University Genomics Core Facility. The FASTX-Toolkit v0.0.13 (http://hannonlab.cshl.edu/fastx_toolkit/) was used to remove adapter sequences and trim read ends using a quality score cutoff of 20. ChIP-seq data were aligned to the hg19 genome assembly (retrieved from http://hgdownload.cse.ucsc.edu/goldenPath/hg19/chromosomes/), using Bowtie 2 v2.0.6 (ref. 63), discarding reads with at least one mismatch and reporting the best alignment, if multiple alignments were present. PCR duplicates were removed using SAMtools[64]. Peaks were detected with MACS v1.4 (ref. 65), using an aligned input DNA sample as control with a threshold for significant enrichment of $P < 1 \times 10^{-9}$. Peaks lists were filtered to remove all peaks overlapping ENCODE blacklisted regions (https://sites.google.com/site/anshulkundaje/projects/blacklists). Wiggle tracks stepped at 25 bp were generated by MACS, normalized by the mean genome-wide wiggle track signal in each dataset and visualized on the Integrative Genomics Viewer[66].

**Identification of variant enhancer loci.** For each CRC sample, we combined the H3K27ac peaks called in that sample with peaks from four normal colon crypt epithelium samples (C28, C29, C37, Crypt5) and merged overlapping peaks using BEDTools[67]. The R package DESeq[12] was used to identify peaks from the merged list with differential enrichment of H3K27ac in the CRC sample relative to four crypt samples at multi-test corrected (Benjamini–Hochberg) $P < 0.05$. Master lists of all unique gained or lost VELs found across the CRC cell lines were created by combining the DESeq-based VEL calls for each CRC line and merging all overlapping loci.

**Quantification of VEL recurrence.** Recurrence was assessed from the master VEL lists, with a VEL considered present in a given cell line if at least one VEL call for that CRC sample overlapped the collapsed VEL coordinates. We tabulated these results with a 1 or 0 for presence or absence, respectively, producing a matrix of dimensions (number of CRC cell lines) × (number of master VELs). The matrices for gained and lost VELs are provided as Supplementary Data 2 and 3, respectively. We used permutation analysis of these matrices to determine the expected, random distribution of recurrence values and calculate the $P$ value and FDR for each level of recurrence (1–31 out of 31 CRC lines).

Super-enhancers were called using the ROSE package[36]. To determine poised/primed status underlying gained VELs, coordinates of gained VELs from the master list were intersected with H3K4me1 and H3K27me3 MACS-called peaks in normal colon crypt and the number of VELs overlapping each mark counted.

**Saturation analysis.** Saturation analyses were performed separately for gained and lost VELs. First, subsets of the 31 CRC cell lines in our panel were drawn at random; 5000 randomizations were performed for each n except for subset sizes with <5,000 possible combinations ($n = 1$–$3$ and $28$–$30$ CRC lines), in which case all combinations were tested. For each random subset, DESeq VEL calls for the subsetted CRC lines were combined, overlapping loci were collapsed, and the total number of unique VEL calls was counted. Saturation curves were fit to this data using Prism (Version 5.0a) to calculate the theoretical maximum number of gained VELs—152,588 (95% confidence interval 152,196–152,979)—and lost VELs—47,460 (95% c.i. 47,384–47,535). Thus, the 121,806 gained VELs identified in our panel represent ~80% of all possible gained VELs (95% c.i. 79.6–80.0%), and the 42,174 lost VELs called in our panel represent ~89% of all possible lost VELs (95% c.i. 88.7–89.0%).

**Transcriptome studies.** Transcript levels were quantified using Affymetrix Human Exon 1.0 ST exon arrays, as previously described[3], for five normal colon crypt samples and 22 of the 31 CRC cell lines, as well as tissue specimens from sixteen normal colon samples and 120 primary and metastatic colorectal tumours.

**Prediction of VEL target genes.** Interactions between genes and enhancers were predicted using the PreSTIGE algorithm[13–15] which identifies distance-restricted enhancer–gene pairs for which both H3K4me1 enrichment and transcription levels are specific to the tissue of interest. High-stringency PreSTIGE predictions were generated for all CRC lines for which both H3K4me1 ChIP-seq profiling and microarray expression data were available. PreSTIGE predictions for crypt specimens were generated using median expression from the five crypt samples included in the microarray data. All normal crypt and CRC predictions were then concatenated to create a master file of all colon-specific enhancer–gene pair predictions. VEL coordinates were intersected using BedTools with this file to assign putative gene targets. PreSTIGE predicted target genes of recurrent gained (G10 +) and lost (L14 +) VELs are listed in Supplementary Data 7 and 8.

**Correlation of VELs and target gene expression.** For each of the 22 CRC lines with expression data, genes were grouped based on the number of associated gained or lost VELs, removing genes associated with both VEL types and genes that were expressed at or below background noise levels in either the CRC line or normal crypt samples. Expressed genes not associated with any VELs were assigned to the nonVEL control set. MWW tests were used to determine statistical differences of gene sets versus the control (nonVEL) genes. Data was boxplotted in R,

removing outliers. For the analyses presented in Fig. 1e, genesets were collapsed across all CRC samples. Significance testing and plotting were performed identically for gene sets defined by the recurrence of associated VELs (Supplementary Fig. 2a). To assess whether recurrent VEL genes were reproducibly dysregulated in patient tumour samples, first, genes dysregulated in the 22 CRC cell lines with microarray expression data were identified, regardless of association with VELs. Genes were classified as overexpressed if they showed more than a 50% increase in expression over normal colon in at least half of the CRC lines, that is, for $\geq 11/22$ CRC lines, (CRC expression)/median(crypt expression) > 1.5 ($n = 1686$ genes). Underexpressed genes were defined as those showing more than a 50% greater expression in normal colon in at least half of the CRC lines, that is, for $\geq 11/22$ CRC lines, (median(crypt expression))/(CRC expression) > 1.5 ($n = 1396$ genes). Next, dysregulated genes were classified based on VEL status into four groups: overexpressed genes not associated with any VELs ($n = 1106$ genes), overexpressed genes associated with recurrent gained VELs (G10 +, $n = 352$), underexpressed genes not associated with any VELs ($n = 418$) and underexpressed genes associated with recurrent lost VELs (L14 +, $n = 589$). Dysregulated recurrent VEL genes, regardless of validation, were visualized in a row-normalized heatmap (Fig. 2d). Finally, the number of genes in each group that were similarly misexpressed in the 120 primary CRC samples was determined to calculate a validation rate. Misexpression in the microarray data for 120 primary CRC samples was defined analogously to the CRC cell lines, such that for overexpressed genes, (primary CRC expression)/median(normal tissue expression) > 1.5 in $\geq 60/120$ samples, and for underexpressed genes, median(normal tissue expression)/(primary CRC expression) > 1.5 in $\geq 60/120$ samples. Significance was calculated by $\chi^2$ tests.

**CRISPR–Cas9-mediated enhancer disruption.** Enhancer editing was performed in collaboration with the Genome Engineering and iPSC Center (GEiC) at Washington University. Guide RNA (gRNA) sequences were designed and then used to target disruption of putative enhancers at the PHLDA1 and MYC loci (Supplementary Data 9). For each target enhancer, we utilized a CRISPR–Cas9-based strategy whereby the enhancers are first cut with gRNA-directed Cas9, and then antibiotic resistance cassettes including an exogenous promoter are inserted at the cut site via non-homologous end joining (NHEJ). HCT116 cells are co-transfected with (1) enhancer-specific gRNA expression plasmids, (2) a linearizing gRNA expression plasmid, (3) a Cas9 expression plasmid, (4) a donor plasmid containing PGK-driven puromycin resistance gene (PuroR) flanked by the linearizing gRNA target sites and (5) a donor plasmid containing PGK-driven blasticidin (BlastR) resistance gene flanked by linearizing gRNA target sites. The co-transfection of these reagents yields blunt-ended, linearized puroR and BlastR 'armless' sequences that are then incorporated via NHEJ in the site cut after cleavage of the genomic DNA by the enhancer-specific gRNA. Pools of edited cells are then selected by sequential drug selection to blasticidin and puromycin. Correctly edited cell pools are then verified by PCR, using primer pairs in which one primer lies either 5′ or 3′ to the endogenous cut site, and the corresponding primer lies within either the blast or puro sequences. All seven enhancers targeted for disruption using this strategy were verified by PCR to contain both targeted blasticidin and puromycin resistance cassettes within the pool of cells. RNA was extracted from CRISPR–Cas-edited cell pools and unedited parental cells via Trizol extraction. Real-time qPCR was performed using TaqMan probes to PHLDA1 and MYC. GAPDH was used as an endogenous control.

**Gene ontology and network analysis.** GO analyses were performed on genes associated with recurrent gained and lost VELs (G10 + and L14 +) using g:Profiler and the Enrichment Map plug-in to Cytoscape[68–70]. $P$ value and FDR cutoffs used were both <0.05.

**Transcription factor enrichment.** TF motif enrichment analysis was performed with the findMotifsGenome tool in the HOMER command line software (http://homer.salk.edu/homer/index.html) using unique gained VELs (gained VELs present in exactly one CRC line) as the background set ( − bg option). For the analysis of experimentally determined TF binding sites in colon, ChIP-seq peak bed files for factors profiled in colon were retrieved from the Cistrome Data Browser. Coordinates were converted from hg38 to hg19 using the UCSC Genome Browser liftOver command line tool (https://genome.ucsc.edu/goldenpath/help/hgTracksHelp.html#Convert). Gained VELs from the master list were intersected with peak coordinates of each factor. Next, for each recurrence level (1–31 CRC lines), we calculated the proportion of all gained VELs with recurrence at or above that number that overlapped each factor. Using linear modelling in R, best fit lines were calculated for this data. To determine which factors showed the greatest increase of binding site enrichment with recurrence, factors were ranked by the slope of their best fit lines.

**Association with CRC GWAS risk loci.** SNPs associated with CRC by GWAS were retrieved from the NHGRI-EBI GWAS Catalogue (http://www.ebi.ac.uk/gwas/) on June 15, 2015 (ref. 37). Linkage disequilibrium (LD) information was pulled from HapMap, and used to identify SNPs in tight LD (LOD > 2, D′ > 0.99) with each GWAS lead SNP in the population in which the SNP was found to be

associated with CRC risk (CEU, JPT or YRI). GWAS SNPs that lacked any LD SNPs were removed, leaving 75 lead SNPs. All analyses were performed with each lead SNP and its set of LD SNPs considered as a unit that is, if a genomic feature overlapped a lead SNP or any of its LD SNPs, then the feature and lead SNP were considered associated. VSE was performed as previously described[3,13,38]. Briefly, 1,000 sets of 75 SNPs were randomly selected such that the number of LD SNPs matched the CRC risk SNP set. The proportion of the random SNPs associated with each genomic feature of interest was calculated, and the distribution of the randoms was used to generate a $P$ value for enrichment (Fig. 3b). Predicted target genes of SNP-associated VELs were determined using PreSTIGE (as described above), or by nearest expressed gene for VELs without PreSTIGE predictions, as well as previous publications[71–75].

**JQ1 experiments.** 2,500–5,000 cells per well were plated in a 96-well plate and the following day were treated in triplicate or quadruplicate with a range of JQ1 concentrations (100–5,000 nM) or vehicle (0.1% DMSO (v/v)). Relative cell number was assessed 72 h later using the CellTiter-Glo assay (Promega, cat. G7572) according to the manufacturer's protocol, and the luminescence was measured on a Wallac Victor$^3$ V 1420 Multilabel Counter and IC$_{50}$ values were calculated from these data. For cell-cycle assays, cells were plated to at $\sim 1 \times 10^6$ cells per well in six-well plates. The following day, cells were treated with 500 nM JQ1 or 0.05% DMSO (v/v) control. After 48 h, cells were harvested, $2 \times 10^6$ cells were resuspended in 1 ml cold PBS. Cells were fixed by adding dropwise to 9 ml cold 70% ethanol, and incubated overnight at $-20\,^\circ$C. Cells were stained with propidium iodide (Life Technologies, cat. P3566) in PBS + 0.1% Triton-X and incubated 30 min at 37 $^\circ$C, then were kept at 4 $^\circ$C until analysed on a Beckman-Coulter Epics XL. For apoptosis assays, cells were plated to at $\sim 1 \times 10^6$ cells per well in 6-well plates. The following day, cells were treated with 500 nM JQ1, 50 nM staurosporine, or 0.05% DMSO (v/v) control. After 48 h, cells were harvested (discarding media and non-adherent cells) and $1 \times 10^6$ cells were stained using the FITC Annexin V Apoptosis Detection Kit I (BD Biosciences, cat. 556547) according to the manufacturer's protocol and promptly analysed on a Beckman-Coulter Epics XL. WinList 3D, and WinList 3D plus ModFit (Verity Software House) were used for apoptosis and cell-cycle analysis, respectively. All assays were run in biological duplicates. For gene expression analyses, cells were plated to at $\sim 0.5 \times 10^6$ cells per well in six-well plates and treated with 500 nM JQ1 for 0.5, 1, 6, or 24 h, or 0 h as a control, following which cells were lysed and RNA extracted with TRIzol (Life Technologies, 15596-026) according to the manufacturer's protocol. RNA-seq was carried out on an Illumina HiSeq, and 100 bp paired-end reads were obtained. Sequences were aligned to the genome using TopHat v1.3.2 and fragments per kilobase per million reads (FPKMs) were calculated using Cufflinks v1.3.0 (refs 76,77). FPKM values were floored to 0.3 and FPKMs were quantile normalized to minimize potential technical artifacts. Only genes actively expressed (defined as more than twice the background signal, FPKM > 0.6) in untreated controls were considered, resulting in 11,253–11,302 genes per CRC line. For each of the six CRC lines tested, genes were grouped based on the number and recurrence of associated gained VELs, removing any genes also associated with recurrent lost VELs. The fold change in expression at each treatment time point compared with control (expression $t_x$/expression $t_0$) was calculated for each gene. MWW tests of each gained VEL gene set against the control set of all expressed genes were performed to determine significance.

**Gene set enrichment analysis.** Genes that were both associated with recurrent gained VELs (G10 + ) present in HCT116 and 1.5-fold overexpressed in HCT116 compared with normal colon were identified. GSEA was performed using the GseaPreranked tool in command line software v2.2.0 (software.broadinstitute.org/gsea/downloads.jsp[78]; to assess enrichment of the HCT116 recurrent VEL gene set in a ranked list of 'fitness' genes in HCT116 from the Toronto KnockOut (TKO) library[50].

**Mouse xenografts.** Tumours were induced by subcutaneous injection of $\sim 0.5$–$10 \times 10^6$ cells into the flank of six to eight week old, female nude mice obtained from the Athymic Animal and Xenograft Core of the Case Comprehensive Cancer Center. CRC lines chosen for xenograft tested negative for mycoplasma and other contaminations. Mice were monitored until tumour size reached $\sim 75$–100 mm$^3$, at which point treatment was initiated. Mice received either 50 mg kg$^{-1}$ JQ1 (or vehicle) by intraperitoneal injection once each day or 20 mg kg$^{-1}$ JQ1 (or vehicle) intraperitoneal injection twice each day for 27 days, with ten mice per group. Post-hoc power analysis indicates that with this sample size and $\alpha$ error = 0.05, effects sizes of 1.3 will be detected with power $(1 - \beta$ error) 0.8. Mouse xenograft studies were performed in accordance with protocols approved by the Case Western Reserve University Institutional Animal Care and Use Committee (Protocol Number: 2013-0179), and in collaboration with the Athymic Animal and Xenograft Core of the Case Comprehensive Cancer Center.

**Data availability.** All sequencing data has been deposited in the GEO database under accession codes GSE77737 (new datasets) and GSE36401 (previously published datasets). The authors declare that all data supporting the findings of this study are available within the article and its Supplementary Information Files or from the corresponding author upon reasonable request.

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

## Acknowledgements

NIH P50CA150964 (Case GI SPORE) (S.D.M. and P.C.S.), R01 CA160356 (P.C.S. and S.D.M.), R01 CA143237 (P.C.S.), R01 CA204279 (P.C.S.), R01 CA193677 (P.C.S.), R01 CA206505 (R.A.K.), F30 CA186633 (J.J.M.), T32 GM007250 (Case MSTP) (J.J.M. and A.J.C.), TL1 RR02499 (Case CTSA) (A.J.C.). Cytometry and Imaging Microscopy, Genomics, and Athymic Animal and Xenograft Cores of the Case Comprehensive Cancer Center (P30 CA043703). Kristen Weber-Bonk for assistance with xenograft studies; Alexander J. Federation for computational assistance.

## Author contributions

Conceptualization: A.J.C., O.C., S.D.M. and P.C.S.; Methodology: A.J.C., O.C., C.F.B., L.B., J.E.B., R.A.K., S.M.P.-M. and P.C.S.; Software/computation: A.J.C., A.S., O.C. and J.M.L.; Formal Analysis: A.J.C., A.S., O.C., J.M.L., S.C.M. and P.C.S.; Experiments/data collection: A.J.C., K.L., C.F.B., G.D., L.B., and S.M.P.-M.; Resources: L.M., M.F.K., J.W., J.E.B., R.A.K. and S.D.M.; Data Curation: A.J.C. and A.S.; Writing—Original Draft: A.J.C. and P.C.S.; Writing—Review & Editing: A.J.C., J.J.M., R.A.K., N.A.B., S.D.M. and P.C.S.; Visualization: A.J.C., A.S.,. J.M.L., S.C.M. and P.C.S.; Supervision: N.A.B. and S.D.M.

## Additional information

Competing financial interests: JEB has ownership interest (including patents) in Tensha Therapeutics and is a consultant/advisory board member of Tensha Therapeutics. Other authors do not report conflicts of interests.

