## [Peer Review File · Nature Communications]

Reviewer #1 (Remarks to the Author)

In this study, the authors identified gained and lost Variant Enhancer Loci (VELs) in CRC tumors and cell lines, followed by characterization of VELs using genetic variation, super enhancers, transcription factors and epigenetic marks. They also explored the relationship between drug sensitivity and VELs. This study demonstrated the importance of chromatin alteration in CRC. And the large-scale enhancer sequencing data they generated would be useful for the community. Overall, the study represents a work of relatively high standard. The methods and data are generally rigorous and supported the conclusions. However, some aspects of this study need further clarification.

Major Remarks:

- 1) Line 120, It looks strange to use DESeq for identifying differential H3K27ac peaks, since DESeq is not a tool specific for Chromatin sequencing. Prior assumption of data distribution (for example, normal or binomial distribution) should be considered.
- 2) Line 133, the target genes of enhancers were predicted computationally. If applicable, the Hi-C data should be used to characterize the enhancers and target genes.
- 3) Line 111, the mean of bar promoter in Fig.1b seems not to be 13632 as claimed in the text. H3K27ac is not a promoter mark. Given that the promoter was defined as 2kb surrounding TSSs (1kb is more suitable), the H3K27ac peaks may be associated with very few promoters when narrowing the promoters to 1kb surrounding TSSs.
- 4) Line 270-273, only 1% of VELs are associated with SNPs, the number is too small, which is in contradiction with the majority of risk loci reside in putative H3K27ac-enhancers in crypts or CRC, please explain.

Minor Remarks:

- 2) Line 139-140, I could not find data to support the sentence, how to obtain 89% and 80%?
- 3) Line 144, were expression data used to do correlation plot? Not clear what the enhancer profile indicates?
- 4) Line 174-200, were the recurrent VEL genes defined by $P < 0.001$? The genes used in Figure d,e were not clearly defined.
- 5) In Fig.3b, why the mean values of all bars are the same?

Reviewer #2 (Remarks to the Author)

This manuscript ascertains enhancer sites across the epigenome based on histone mono-methylation marks of histone H3 (H3K4me1 and H3K27ac) in colorectal cancers, with attempts to show relevance of the enhancer sites based on current information or structure. The authors combined 42 datasets to get a high resolution view of H3K27ac profiles, and utilized 7 fresh normal colonic crypts and 31 diverse colorectal cell lines and 4 primary tumors. For H3K4me1, the authors used normal crypts plus a "subset" of cell lines, with the cell lines for discovery and tissues for verification. There was 82% overlap between H3K27ac and H3K4me1 sites, and up to 64% were contained in open chromatin. The majority of the 33,561 H3K27ac peaks were located distal to promoter sites, and contained signature features of enhancer elements. Relative to normal

crypts, the distal sites may be more particularly prone to aberrant changes in cancer, using PreSTIGE software prediction. The authors state that "early stage tumors generally preserve a considerable portion of the normal enhancer epigenome...compared to late stage tumors that have undergone a more dramatic shift". Interestingly, this data did not correlate with MSI status, which most commonly is the result of hypermethylation and the CIMP genotype.

The authors argue that many of the enhancer sites are known to activate known and novel oncogenes, with most part of super enhancer (cluster) sites and are "hotspots". The authors identified 75 methylated enhancer sites in cancer relative to normal crypts (and with the associated genes generally overexpressed, such as MYC, BMP4, PHLDA1, SOX9, FOXQ1 and TRIB3), and 67 sites that were more methylated in normal crypts (and with the associated genes generally repressed in expression, such as E2F2 and SIRT6). Adenomas were not examined, but the authors state that "...these sites are acquired at the early stages of malignant transformation and may be essential for maintenance of the malignant phenotype". Up to 44% of sites more methylated in normal crypts were constituents of super enhancer sites, while 96% of sites more methylated in cancer were parts of super enhancers (e.g. MYC, ETS2, CDX2, RAD21, SMC1A, SMC3, NIPBL, JUND, JUN, ATF2). Additionally, about half of GWAS loci map to acquired enhancers. Most of the enhancers originate from primed chromatin, suggesting a reawakening of developmental enhancers in cancer. Pharmacologic inhibition (with JQ1, and inhibitor of BRD4) or functional knock out of 17,661 genes in HCT116 cells activated by enhancers mitigate colorectal cancer cell growth in cell lines and xenografts.

Strengths:

1. Well written, well done experimentally.
2. Show some (limited) functional data that support the author's hypothesis

Weaknesses:

1. Smaller sample sizes have been previously examined, and the findings are not novel (e.g. Genome Biol. 2014 Sep 20;15(9):469; Genome Res. 2015 Apr;25(4):467-77; Nat Commun. 2014 Sep 30;5:5114; Clin Proteomics. 2014 Jun 3;11(1):24). The current manuscript extends those prior findings to a larger sample size, and reinforces the concept of super enhancer sites.
2. Data heavily reliant on H3K27ac marks versus H3K4me1 marks. Also, Fig 2H shows H3K27me3 marks. Does this add anything to the discovery or validation sets over H3K27ac?
3. Adenomas, neoplasms that are still benign but intermediary in the progression sequence, would have been ideal to show enhancer epigenome changes as a presumed intermediate step, in addition to "early" and "late" stage tumors. Including adenomas would make this paper more unique.
4. Can the authors explain the difference between hypermethylation in CIMP/MSI sporadic colorectal cancers, and this hypermethylation?
5. The authors often use words such as "a significantly high proportion" or "frequently" instead of providing exact proportions or numbers. Some are available in tables/figures, but others are totally descriptive.

Reviewer #3 (Remarks to the Author)

In this work, Cohen et al. studied the enhancer variation or Variant Enhancer Loci (VELs) in a large collection of colorectal cancer cell lines, primary tumors and normal colon samples, based mainly on ChIP-seq data for H3K27ac. They show that gained VELs result mainly from primed enhancers in normal cells. Moreover, they provided multiple evidences for the pathological relevance of recurrently gained VELs, including strong enrichment for relevant GWAS SNPs, association to key cancer and fitness-related genes, and increased sensibility of associated genes to JQ1 treatment.

This study represents a comprehensive effort to characterize enhancer variation between normal and cancer cells and strongly enhance our current understanding of epigenetic mechanisms

involved in cancer. However, the relevance of the manuscript would be strengthened by more directly addressing the functional implications of some of the recurrent VELs identified in this study.

Major remarks

1. My main concern is the lack of direct evidence that the gained or loss VELs have functional implication either on the expression of neighbor genes or on cell growth or fitness. The authors should select one or few highly recurrent gained VELs (eventually also overlapping with GWAS SNPs) for CRISPR deletion on selected cell lines.

2. In the same line, the functional effect of SNP variation on enhancer activity of VELs should be addressed experimentally. This could be achieved by reporter assays, CRISPR knockin or by investigating the natural variants within their cell line collection.

3. The motif analysis, indicating enrichment for MYC, AP1 and Cohesin complexes in recurrent VELs (figure 2H), is not very convincing. Overall, there is a general tendency of the factors to be more frequently bound in recurrent VELs. For instance, the AP1 and Cohesin complexes are the most frequently found factors in recurrent VELs but also in non-recurrent VELs, most likely indicating wide binding patterns of these factors. For example, the authors could look for overrepresented factors comparing their rank between recurrent and non-recurrent VELs. Indeed, based on panel 2H, some TF (shown in gray) appears to show this pattern.

4. Figure 4: it will be interesting to analyze whether there is any correlation between the sensitivity of each cell line to the JQ1 treatment and the number of gained/lost VELs and super-enhancers.

Minor remarks

5. Figure 2J and 3E: It is not clear whether the panels show the results for recurrences between 1 and 10. If yes, the legend should be clarified. If not the results should be shown.

6. Figure 3B: results for recurrences between 1 and 10 should be also shown.

7. Figure 3C: The precise positions of the indicated SNPs should be shown. It could be useful to also show a zoomed screenshot around the two SNPs.

8. Typo in Figure 3D: some genes appear in bold without any apparent reason.

Reviewers' comments:

Reviewer #1 (Remarks to the Author):

In this study, the authors identified gained and lost Variant Enhancer Loci (VELs) in CRC tumors and cell lines, followed by characterization of VELs using genetic variation, super enhancers, transcription factors and epigenetic marks. They also explored the relationship between drug sensitivity and VELs. This study demonstrated the importance of chromatin alteration in CRC. And the large-scale enhancer sequencing data they generated would be useful for the community. Overall, the study represents a work of relatively high standard. The methods and data are generally rigorous and supported the conclusions. However, some aspects of this study need further clarification.

We thank the reviewer for recognizing the importance of our findings and acknowledging the quality of our data.

Major Remarks:

1) Line 120, It looks strange to use DESeq for identifying differential H3K27ac peaks, since DESeq is not a tool specific for Chromatin sequencing. Prior assumption of data distribution (for example, normal or binomial distribution) should be considered.

In the original paper describing DESeq (Anders & Huber, Genome Biol 2010), the authors include analysis of ChIP-seq data and demonstrate that this method is suitable for analysis of ChIP-seq data. However, we see the reviewer's point that it is important to validate that the assumptions of the analytical methods hold true for our datasets. As such, we analyzed the distribution of our ChIP-seq data. Our data largely fits a negative binomial distribution, which is the model that DESeq uses. We think that the DESeq analyses performed in our paper are a clear strength of our analysis, because it provides a statistically rigorous method (Benjamini-Hochberg corrected P values of < 0.05) for identifying enhancer differences. Histograms of the ChIP-seq data serving as DESeq input (read count per called enhancer) are shown below for four representative CRC cell lines; overlaid red, dashed lines represent best-fit negative binomial distributions.

2) Line 133, the target genes of enhancers were predicted computationally. If applicable, the Hi-C data should be used to characterize the enhancers and target genes.

To address this concern, we compared our enhancer-gene predictions to TAD structure defined in other cell types (Dixon, *et al.*, Nature 2015). TAD boundaries are conserved across cell types and are believed to constrain enhancer-gene interactions. Consistent with this, we find that >87% of our enhancer-gene predictions are contained within TAD boundaries.

Our method for enhancer-gene assignment is based on a published method called PreSTIGE (Corradin, *et al.*, Gen. Res 2014), which was extensively validated using both 3C and Hi-C-defined enhancer-gene interactions. While there have been capture Hi-C experiments done on some available CRC cell lines, we're not aware of any genome-wide Hi-C data that could be utilized to verify our enhancer-gene predictions. This would require Hi-C studies at very high resolution (< 5 kb) in the cell line models used in our study. This level of resolution requires very deep sequencing (> 2 billion reads per sample) and as such goes far beyond the scope of the present study.

3) Line 111, the mean of bar promoter in Fig.1b seems not to be 13632 as claimed in the text. H3K27ac is not a promoter mark. Given that the promoter was defined as 2kb surrounding TSSs (1kb is more suitable), the H3K27ac peaks may be associated with very few promoters when narrowing the promoters to 1kb surrounding TSSs.

We apologize for the confusion. The figure displays the mean, whereas the text previously reported the median. We have clarified this in the text.

Respectfully, the reviewer is mistaken in stating that H3K27ac is not a promoter mark. There are multiple lines of evidence from both ENCODE and Epigenomics Roadmap consortia supporting H3K27ac as a mark at both active promoters and enhancers. In any event, we performed the analysis that the reviewer suggests. As indicated in the plots below, there is very little difference (1-5%) in the number of H3K27ac peaks that are classified as putative promoters/enhancers. We now state this result in the text.

4) Line 270-273, only 1% of VELs are associated with SNPs, the number is too small, which is in contradiction with the majority of risk loci reside in putative H3K27ac-enhancers in crypts or CRC, please explain.

We think this is point of clarification. The statement “the majority of risk loci ... reside in a putative H3K27ac-enhancer in crypts or CRC” and the percentages presented in Fig 3A come from an assessment of the proportion of all CRC risk loci that co-localize with a H3K27ac peak. For our analyses, a CRC risk locus is defined as a SNP associated with CRC predisposition by GWAS (the lead SNP) plus SNPs in tight linkage disequilibrium with the lead SNP (LD SNPs; see Methods for further details). Out of the total of 75 CRC risk loci, 57 (76%) overlap an H3K27ac peak in either normal colon crypt samples or in CRC cell lines. In addition, we determined that 38 (50.7%; blue and red sections in Fig 3A) of the 75 risk loci overlap one or more recurrent gained VELs.

We also calculated the converse, i.e. the percentage of VELs that overlap with CRC risk loci at various levels of VEL recurrence. These results are displayed in Fig 3E. For example, there are 15,065 gained VELs that are present in 10 or more of the 31 CRC lines (recurrent gained VELs; G10+ in Fig 3E). Of these, 54 (0.36%) overlap a CRC risk locus. It should also be noted that a single risk locus can overlap more than one recurrent gained VEL; for example, these 54 recurrent gained VELs are associated with the 38 CRC risk loci described above.

Because the number of enhancers or VELs is so much greater than the number CRC risk loci, the overlap of the two groups contains a majority of all risk loci, but only a very small percentage of enhancers/VELs:

Minor Remarks:

2) Line 139-140, I could not find data to support the sentence, how to obtain 89% and 80%?

Saturation analysis (Fig 1g) predicts that if infinite samples were profiled, the total number of gained VELs identified would be 152,588 (95% confidence interval: 152,196-152,979). We have identified 121,806 gained VELs in our panel, or ~80% of all possible gained VELs (95% c.i. 79.6-80.0%). Similarly, saturation analysis predicts a total of 47,460 possible lost VELs (95% c.i. 47,384-47,535). Therefore, the 42,174 lost VELs called in our panel represent ~89% of all possible lost VELs (95% c.i. 88.7-89.0%). We have updated the methods to include these numbers.

3) Line 144, were expression data used to do correlation plot? Not clear what the enhancer profile indicates?

The hierarchically clustered heatmap in Fig 1l uses H3K27ac ChIP-seq data (peak RPKMs for 20,000 randomly selected putative enhancer elements). Spearman correlations were calculated across the 20,000 peak RPKMs for all pairs of normal colon crypt samples and CRC lines. These pairwise correlations were then used for hierarchical clustering and plotted in the heatmap shown in Fig 1l. We've clarified this in the text.

4) Line 174-200, were the recurrent VEL genes defined by $P < 0.001$? The genes used in Figure d,e were not clearly defined.

The P value is that associated with the recurrence of the VEL, not the gene. Recurrent VEL genes are the predicted gene targets of recurrent VELs. The genes used in Figure 2G (formerly 2E) are now listed in Supplemental Tables 7 and 8.

5) In Fig.3b, why the mean values of all bars are the same?

Fig. 3b displays the results of a previously published method called variant set enrichment (VSE) analysis. When the VSE data is plotted, the enrichment score calculated from the real data (represented by the black and red diamonds) and the 1000 randomizations (represented by the gray boxplots) are normalized (Cowper-Saï-lari, *et al.*, Nat Genet 2012; Akhtar-Zaidi, *et al.*, Science 2012). The resulting "relative enrichment score" is directly comparable between the different sets of genomic features, allowing visual comparison of the data sets. The normalization is performed by centering the data around the median of the randomizations and then scaling the data based on the variance of the randomizations. The median centering produces the alignment of the median bars the reviewer noted.

Reviewer #2 (Remarks to the Author):

This manuscript ascertains enhancer sites across the epigenome based on histone mono-methylation marks of histone H3 (H3K4me1 and H3K27ac) in colorectal cancers, with attempts to show relevance of the enhancer sites based on current information or structure. The authors combined 42 datasets to get a high resolution view of H3K27ac profiles, and utilized 7 fresh normal colonic crypts and 31 diverse colorectal cell lines and 4 primary tumors. For H3K4me1, the authors used normal crypts plus a "subset" of cell lines, with the cell lines for discovery and tissues for verification. There was 82% overlap between H3K27ac and H3K4me1 sites, and up to 64% were contained in open chromatin. The majority of the 33,561 H3K27ac peaks were located distal to promoter sites, and contained signature features of enhancer elements. Relative to normal crypts, the distal sites may be more particularly prone to aberrant changes in cancer, using PreSTIGE software prediction. The authors state that "early stage tumors generally preserve a considerable portion of the normal enhancer epigenome...compared to late stage tumors that have undergone a more dramatic shift". Interestingly, this data did not correlate with MSI status, which most commonly is the result of hypermethylation and the CIMP genotype.

The authors argue that many of the enhancer sites are known to activate known and novel oncogenes, with most part of super enhancer (cluster) sites and are "hotspots". The authors identified 75 methylated enhancer sites in cancer relative to normal crypts (and with the associated genes generally overexpressed, such as MYC, BMP4, PHLDA1, SOX9, FOXQ1 and TRIB3), and 67 sites that were more methylated in normal crypts (and with the associated genes generally repressed in expression, such as E2F2 and SIRT6). Adenomas were not examined, but the authors state that "...these sites are acquired at the early stages of malignant transformation and may be essential for maintenance of the malignant phenotype". Up to 44% of sites more methylated in normal crypts were constituents of super enhancer sites, while 96% of sites more methylated in cancer were parts of super enhancers (e.g. MYC, ETS2, CDX2, RAD21, SMC1A, SMC3, NIPBL, JUND, JUN, ATF2). Additionally, about half of GWAS loci map to acquired enhancers. Most of the enhancers originate from primed chromatin, suggesting a reawakening of developmental enhancers in cancer. Pharmacologic inhibition (with JQ1, and inhibitor of BRD4) or functional knock out of 17,661 genes in HCT116 cells activated by enhancers mitigate colorectal cancer cell growth in cell lines and xenografts.

Strengths:

1. Well written, well done experimentally.
2. Show some (limited) functional data that support the author's hypothesis

We thank the reviewer for noting these strengths.

Weaknesses:

1. Smaller sample sizes have been previously examined, and the findings are not novel (e.g. *Genome Biol.* 2014 Sep 20;15(9):469; *Genome Res.* 2015 Apr;25(4):467-77; *Nat Commun.* 2014 Sep 30;5:5114; *Clin Proteomics.* 2014 Jun 3;11(1):24). The current manuscript extends those prior findings to a larger sample size, and reinforces the concept of super enhancer sites.

As the reviewer notes, previous studies including our own (Akhtar-Zaidi et al, *Science* 2012) have demonstrated the existence of cancer-specific gains and losses of enhancer activity in CRC, as well as enrichment of GWAS SNPs in colon regulatory elements. The papers the review refers to are among those that establish these points, and are indeed part of the foundation of the current work. However, other studies (like the first two the reviewer notes) have primarily focused on DNA methylation, rather than directly assessing marks of enhancer activity via genome-wide ChIP-seq (K4me1 and K27ac). Second, in our view, our study is the first to characterize enhancer alterations in CRC to near saturation. We're aware of only one other study that has successfully presented an analysis of a cancer to this depth, which was performed in medulloblastoma (Lin, *et al.*, *Nature* 2016). The strength of a larger sample size is not simply a confirmation of previous results. It is necessary to enable discovery of recurrent enhancer events that, analogous to recurrent mutational events, are likely to be most relevant to CRC pathogenesis. The larger sample size also enables us to assess to what extent CRC GWAS hits map to enhancers and in doing so, we stumbled across the novel finding that these GWAS SNPs not only lie in enhancers, but these SNPs are specifically enriched in the recurrent VELs and that these enhancers are CRC-specific and not found in normal colon. We think this is particularly novel in that it suggests that the effects of these SNPs may not manifest in normal colon. This has profound implications for how we assess the function of these non-coding variants.

2. Data heavily reliant on H3K27ac marks versus H3K4me1 marks. Also, Fig 2H shows H3K27me3 marks. Does this add anything to the discovery or validation sets over H3K27ac?

We performed ChIP-seq profiling of both H3K27ac and H3K4me1, since these 2 marks are generally considered signatures of active gene enhancers (Rada-Iglesias, et al., *Nature* 2009; Creyton, et al., *PNAS* 2010; Zentner et al., *Genome Research* 2011; etc). While we defined VELs based on H3K27ac, we show there's a strong association between K27ac change and concordant changes in H3K4me1 and DNase hypersensitivity (Fig 1E & Suppl Fig 1). We could have done the analysis focusing first on H3K4me1 differences, these would likely represent a mixture of both poised and active enhancers,

which could confound downstream analyses. In any event, we are making both of these datasets publicly available for the community to analyses in various ways.

With respect to the H3K27me3 data in Fig 5 (formerly Fig 2I, J), the results indicate that the majority of recurrent gained VELs arise from loci that were poised in normal colon epithelium, whereas less than a quarter of unique gained VEL loci display these marks in normal colon. This clear bias is important because it suggests aberrant activation of poised enhancers is a potential mechanism for recurrent gain of enhancers in CRC.

3. Adenomas, neoplasms that are still benign but intermediary in the progression sequence, would have been ideal to show enhancer epigenome changes as a presumed intermediate step, in addition to "early" and "late" stage tumors. Including adenomas would make this paper more unique.

We agree and thank the reviewer for his or her suggestion. We have now added enhancer histone H3K27ac ChIP-seq analysis of 2 adenoma specimens directly obtained from a patient with familial adenomatous polyposis (FAP). The results indicate that during the stepwise progression of CRC, the crypt to early adenoma transition is accompanied by acquisition of a subset of the recurrent gain VEL signature, and that the remaining signature VELs likely arise later, during the adenoma to carcinoma transition. We believe this finding will be of considerable interest to the field, and as such we present the data in an entirely new figure (now Figure 4).

4. Can the authors explain the difference between hypermethylation in CIMP/MSI sporadic colorectal cancers, and this hypermethylation?

While CIMP+ and CIMP- CRC tumors may be distinct at the level of DNA methylation, we did not detect global differences in the VEL profiles of MSI and MSS tumors. If DNA hypermethylation were functionally linked with enhancer histone marks, we would predict that the enhancer and VEL profiles of CIMP+/MSI CRCs would be markedly distinct from the VEL profiles of CIMP-/MSS tumors. We found that unsupervised clustering based on H3K27ac signal across the epigenome shows only nominal differences between MSI and MSS CRCs (Fig 1I), and clustering based on H3K27ac signal at VELs, specifically, shows no segregation of MSI and MSS tumors (Z-test of proportions $P \cong 0.26$). Thus, the changes of enhancer marks observed in CRC are not tightly associated with the DNA methylation changes characteristic of CIMP tumors.

5. The authors often use words such as "a significantly high proportion" or "frequently" instead of providing exact proportions or numbers. Some are available in tables/figures, but others are totally descriptive.

Several instances of language like this were edited to be more specific.

Reviewer #3 (Remarks to the Author):

In this work, Cohen et al. studied the enhancer variation or Variant Enhancer Loci (VELs) in a large collection of colorectal cancer cell lines, primary tumors and normal colon samples, based mainly on ChIP-seq data for H3K27ac. They show that gained VELs result mainly from primed enhancers in normal cells. Moreover, they provided multiple evidences for the pathological relevance of recurrently gained VELs, including strong enrichment for relevant GWAS SNPs, association to key cancer and fitness-related genes, and increased sensibility of associated genes to JQ1 treatment.

This study represents a comprehensive effort to characterize enhancer variation between normal and cancer cells and strongly enhance our current understanding of epigenetic mechanisms involved in cancer. However, the relevance of the manuscript would be strengthened by more directly addressing

the functional implications of some of the recurrent VELs identified in this study.

Major remarks

1. My main concern is the lack of direct evidence that the gained or loss VELs have functional implication either on the expression of neighbor genes or on cell growth or fitness. The authors should select one or few highly recurrent gained VELs (eventually also overlapping with GWAS SNPs) for CRISPR deletion on selected cell lines.

In response, we performed CRISPR-Cas9-mediated disruption of 3 recurrent VELs at the PHLDA1 locus. Additionally 4 sites at the MYC locus were edited as a negative control. The results (now shown in Fig 2E, F and Suppl. Fig 2B, C) provide direct evidence that the 3 enhancers at the PHLDA1 locus regulate PHLDA1 expression.

2. In the same line, the functional effect of SNP variation on enhancer activity of VELs should be addressed experimentally. This could be achieved by reporter assays, CRISPR knockin or by investigating the natural variants within their cell line collection.

As the reviewer suggests, we sought to determine the effects of GWAS risk loci on VEL function by using the sequence variation across a population of CRC tumors. Our set of CRC cell lines is not large enough to provide sufficient statistical power for eQTL or similar analyses. We therefore made use of the largest panel of matched SNP profiles and expression data in CRC that we are aware of, the COADREAD dataset from TCGA, and analyzed the impact of genotype at CRC risk loci that underlie recurrent gained VELs on expression of the predicted VEL gene. We found that tumors from individuals who inherit the risk allele at the gained VEL-associated risk locus, rs6983267, have significantly higher MYC expression ($P < 0.0001$; see image below).

This confirms the results presented in other studies (using different datasets) (Tuupanen, *et al.*, Nat Genet. 2009; Pomerantz, *et al.*, Nat Genet 2009; Yao, *et al.*, Nat Commun 2014).

Several other loci assessed showed borderline significance in our analysis, suggesting that a number of additional gained VEL-associated GWAS risk loci likely have similar, though more moderate, cancer context-dependent effects on oncogene expression. However, even the TCGA COADREAD dataset is small compared to those typically used for eQTL-type analyses. As a result, we are underpowered to adequately test these loci, and we therefore decided not to include this analysis in the paper.

3. The motif analysis, indicating enrichment for MYC, AP1 and Cohesin complexes in recurrent VELs

(figure 2H), is not very convincing. Overall, there is a general tendency of the factors to be more frequently bound in recurrent VELs. For instance, the AP1 and Cohesin complexes are the most frequently found factors in recurrent VELs but also in non-recurrent VELs, most likely indicating wide binding patterns of these factors. For example, the authors could look for overrepresented factors comparing their rank between recurrent and non-recurrent VELs. Indeed, based on panel 2H, some TF (shown in gray) appears to show this pattern.

We apologize for any lack of clarity. We have reworked this section, and now include an analysis of motifs that are enriched (via computational prediction) in recurrent gained VELs over non-recurrent (unique) VELs. This is now coupled with the prior analysis of TF binding sites mapped experimentally through ChIP-seq studies in CRC. We clarify that we are describing the top ranked TFs based on the increase in TFBS enrichment with recurrence (i.e, the slope of the linear regression line). As the reviewer notes, this analysis does not directly assess whether the increase of TFBS enrichment with recurrence is significantly greater for the top ranked factors than for the other factors in the set, and it was not our intention to imply that this is the case. Our main point was to show that among the top ranked factors are several with known roles in CRC and enhancer biology (i.e, AP1, cohesin, etc.). The text has been edited to clarify that the filtering in this analysis was based on rank, and not statistically significant difference, compared to the other TFs.

4. Figure 4: it will be interesting to analyze whether there is any correlation between the sensitivity of each cell line to the JQ1 treatment and the number of gained/lost VELs and super-enhancers.

We thank the reviewer for this suggestion. There is not a significant correlation between JQ1 sensitivity and number of gained, lost, or total VELs ($p > 0.39$). The same is true when only recurrent VELs ($p > 0.50$), highly recurrent VELs ($p > 0.26$), or super enhancers ($p \sim 0.367$) are considered.

We performed a variety of analyses with the goal of identifying enhancer elements that distinguish the most and least responsive CRC cell lines. While we were able to find enhancer sets which appeared to correlate with JQ1 sensitivity, our current panel size did not provide sufficient power to both identify and validate these enhancer sets with statistical significance, and therefore these analyses were not discussed in the manuscript.

Minor remarks

5. Figure 2J and 3E: It is not clear whether the panels show the results for recurrences between 1 and 10. If yes, the legend should be clarified. If not the results should be shown.

Results for non-recurrent gained VELs (present in 1-9 CRC lines) have been added to Fig 5A (formerly Fig 2J). Results for non-recurrent gained VELs and lost VELs (identified in 1-13 CRC lines) have been added to Fig 3E.

6. Figure 3B: results for recurrences between 1 and 10 should be also shown.

Results for non-recurrent gained VELs (present in 1-9 CRC lines) have been added to Fig 3B.

7. Figure 3C: The precise positions of the indicated SNPs should be shown. It could be useful to also show a zoomed screenshot around the two SNPs.

GWAS identified lead SNP positions have been marked with arrowheads in Fig 3C.

8. Typo in Figure 3D: some genes appear in bold without any apparent reason.

We were not able to replicate this error in our versions of Adobe Acrobat or Adobe Illustrator.

Reviewer #1 (Remarks to the Author)

I am satisfied with the revision.

Reviewer #2 (Remarks to the Author)

The authors have revised the manuscript, which reads much better and is clearer. The addition of my suggested role for data from adenomas strengthens the manuscript. It should be noted that since the adenomas are from a Familial Adenomatous Polyposis patient, the issue of methylation (i.e. CIMP or sporadic MSI from MLH1 hypermethylation) would not be addressed. Lastly, the authors' findings as compared to prior studies is still an incremental advance, even though the authors state that this paper is "near saturation" and novel. I wonder if another group come up with >90% saturation versus the high 80% saturation if it would be accepted as a "novel" manuscript. However, the paper is thorough, now follows enhancer data from normal to adenomas to cancer, and is a nice tour of force of information on colorectal cancer development.

Reviewer #3 (Remarks to the Author)

This referee is satisfied by the revised version of the manuscript.

Minor remarks:

- 1) The new analysis concerning motif enrichment should be shown (refer to comments 3)
- 2) I do not see the arrowheads indicating the position of lead SNPs (refer to comments 7)
- 3) The authors might consider showing or mentioning the association between rs6983267 and MYC expression (refer to comments 2)

We would like to thank all three reviewers for their time and constructive feedback.

Reviewer #1 (Remarks to the Author):

I am satisfied with the revision.

Reviewer #2 (Remarks to the Author):

The authors have revised the manuscript, which reads much better and is clearer. The addition of my suggested role for data from adenomas strengthens the manuscript. It should be noted that since the adenomas are from a Familial Adenomatous Polyposis patient, the issue of methylation (i.e. CIMP or sporadic MSI from MLH1 hypermethylation) would not be addressed. Lastly, the authors' findings as compared to prior studies is still an incremental advance, even though the authors state that this paper is "near saturation" and novel. I wonder if another group come up with >90% saturation versus the high 80% saturation if it would be accepted as a "novel" manuscript. However, the paper is thorough, now follows enhancer data from normal to adenomas to cancer, and is a nice tour of force of information on colorectal cancer development.

We acknowledge that we are not directly addressing the relationship between enhancer dysregulation and DNA hypermethylation. While we agree that this is an interesting question, it was not our intention to address it in this manuscript. We would argue that it is a question that merits a full study on its own and, as you kindly noted, the current manuscript already encompasses a large amount of information in terms of new data and findings.

Reviewer #3 (Remarks to the Author):

This referee is satisfied by the revised version of the manuscript.

Minor remarks:

- 1) The new analysis concerning motif enrichment should be shown (refer to comments 3)
The output of the new HOMER motif enrichment analysis has been included as Supplementary Data 4.
- 2) I do not see the arrowheads indicating the position of lead SNPs (refer to comments 7)
The row of the figure identifying SNP locations has been labeled in Fig 3C, and additional explanation has been added to the legend to clarify where SNPs are represented within the figure.
- 3) The authors might consider showing or mentioning the association between rs6983267 and MYC expression (refer to comments 2)
Since this result simply replicates previous findings, we have chosen not to include it in the main manuscript. However, since we will be participating in the journal's transparent peer review system and reviewer comments and our responses will be published, this information will be available to those who are interested.